# Uniform thin ice on ultraflat graphene for high-resolution cryo-EM

Liming Zheng[1,2,11], Nan Liu [3,11] ✉, Xiaoyin Gao[1,11], Wenqing Zhu [4,11], Kun Liu[5,6], Cang Wu [6], Rui Yan[2], Jincan Zhang[1,7], Xin Gao[1,7], Yating Yao[1], Bing Deng [1], Jie Xu[3,8], Ye Lu[3,8], Zhongmin Liu[6], Mengsen Li[5], Xiaoding Wei [4,9,10] ✉, Hong-Wei Wang [3,8] ✉ & Hailin Peng [1,2,7] ✉

Cryo-electron microscopy (cryo-EM) visualizes the atomic structure of macromolecules that are embedded in vitrified thin ice at their close-to-native state. However, the homogeneity of ice thickness, a key factor to ensure high image quality, is poorly controlled during specimen preparation and has become one of the main challenges for high-resolution cryo-EM. Here we found that the uniformity of thin ice relies on the surface flatness of the supporting film, and developed a method to use ultraflat graphene (UFG) as the support for cryo-EM specimen preparation to achieve better control of vitreous ice thickness. We show that the uniform thin ice on UFG improves the image quality of vitrified specimens. Using such a method we successfully determined the three-dimensional structures of hemoglobin (64 kDa), α-fetoprotein (67 kDa) with no symmetry, and streptavidin (52 kDa) at a resolution of 3.5 Å, 2.6 Å and 2.2 Å, respectively. Furthermore, our results demonstrate the potential of UFG for the fields of cryo-electron tomography and structure-based drug discovery.

Cryo-electron microscopy (cryo-EM) has become a major tool in structural biology. As the resolution achievable by cryo-EM improves, the production of the uniformly thin ice required for high-resolution structural determination becomes increasingly important[1–5]. For small biomolecules with molecular weight less than 100 kDa, low contrast in cryo-EM hinders successful reconstruction and lowers the resolution able to be achieved[6,7], given that high-resolution structural determination requires thin ice to minimize the background noise. During the standard process of preparing thin ice in cryo-EM specimen preparation, a thin liquid film is normally obtained by wicking excess solution from a supporting film.

The rough and heterogeneous liquid–solid interfaces during wicking have recently been found to be a fundamental limitation to the production of a reproducible and uniform ice thickness[8]. In 1990 the homogeneity and thickness of thin liquid film were found to be markedly influenced by the roughness of the underlying support. That is, the thinner the liquid film, the more dominant the effect of support roughness[9,10]. Thus, the production of uniformly thin vitreous ice would appear to be dependent on the development of an ultraflat supporting film.

To our knowledge, the relationship between the surface flatness of the supporting film and the uniformity of the ice thickness is still poorly

[1]Center for Nanochemistry, Beijing Science and Engineering Center for Nanocarbons, Beijing National Laboratory for Molecular Sciences, College of Chemistry and Molecular Engineering, Peking University, Beijing, China. [2]Beijing Graphene Institute (BGI), Beijing, China. [3]Ministry of Education Key Laboratory of Protein Sciences, Beijing Frontier Research Center for Biological Structures, Beijing Advanced Innovation Center for Structural Biology, School of Life Sciences, Tsinghua University, Beijing, China. [4]State Key Laboratory for Turbulence and Complex System, Department of Mechanics and Engineering Science, College of Engineering, Peking University, Beijing, China. [5]Hainan Provincial Key Laboratory of Carcinogenesis and Intervention, Hainan Medical College, Haikou, China. [6]Department of Biology, School of Life Sciences, Southern University of Science and Technology, Shenzhen, China. [7]Academy for Advanced Interdisciplinary Studies, Peking University, Beijing, China. [8]Tsinghua-Peking Joint Center for Life Sciences, Tsinghua University, Beijing, China. [9]Beijing Innovation Center for Engineering Science and Advanced Technology, Peking University, Beijing, China. [10]Peking University Nanchang Innovation Institute, Nanchang, China. [11]These authors contributed equally: Liming Zheng, Nan Liu, Xiaoyin Gao, Wenqing Zhu. ✉e-mail: nanliuem@tsinghua.edu.cn; xdwei@pku.edu.cn; hongweiwang@tsinghua.edu.cn; hlpeng@pku.edu.cn

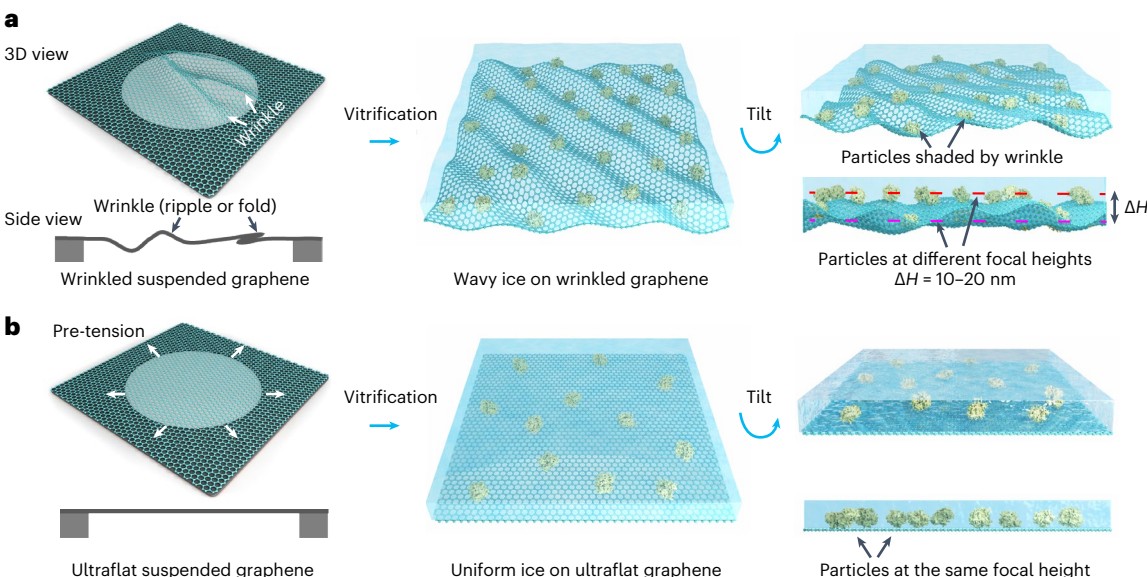

**Fig. 1 | Graphene roughness dominates the uniformity of ice thickness.**
**a**, Schematic illustration of wrinkled rough graphene resulting in wavy ice with inconsistent ice thickness, which can result in a focal height difference ($\Delta H$) of 10–20 nm. The protein particles are shaded by the wrinkled graphene during cryo-ET imaging. Note that the wrinkles in graphene include the ripples and folds, as shown in the side view of wrinkled suspended graphene. **b**, Schematic illustration of pre-tensioned UFG resulting in a uniform ice layer, with proteins adsorbed on the UFG at the same height.

understood, and an ultraflat supporting film for thin ice deposition has not been created. Dense static wrinkles (folds and ripples) are inevitable even on thin graphene oxide nanosheets and graphene films grown on copper foil, although both have proven to be helpful for better cryo-EM specimen preparation[4,7,11–21]. The height difference of the wrinkled surface can be up to dozens of nanometers (Extended Data Fig. 1), and will directly shape the ice, leading to non-uniform ice thickness and a varying height distribution of particles in the data-collection region (Fig. 1a). Moreover, the wrinkled surface will project obviously strong noise at high tilt angles[22], thus becoming a severe problem when applied in cryo-electron tomography (cryo-ET) (Fig. 1a).

In this Article we present a pre-tensioned ultraflat graphene (UFG) without wrinkling for uniform thin ice preparation (Fig. 1b), in which target particles are adsorbed onto the UFG surface at the same plane, which sequesters them from the air–water interface. During the +60° to −60° tilt of the UFG-prepared specimens, we did not observe any of the corrugated features that are usually present in the conventional rough graphene membrane. Using the UFG we obtained a reconstruction of hemoglobin (64 kDa) at a resolution of 3.5 Å, a reconstruction of α-fetoprotein (67 kDa) at a resolution of 2.6 Å and a reconstruction of streptavidin (52 kDa) at a resolution of 2.2 Å using single-particle cryo-EM, which enables the visualization of structural details at atom-level resolution.

## Results

### Design of ultraflat suspended graphene

Conventional and extensively used graphene films grown on copper foil by chemical vapor deposition (Fig. 2a) are not ideally as flat as one might expect. The graphene flatness is affected by three kinds of broadly observed corrugations[23], that is, dense rolling lines of copper foil (Extended Data Fig. 2), strain-induced step bunches, and wrinkles during graphene growth (Fig. 2b). Such corrugations result in a notice-able height variation of the graphene film and can reach up to dozens of nanometers, even at a lateral scale of several micrometers (Fig. 2c and Extended Data Fig. 2). Consequently, wrinkles are inevitably printed onto the holey substrate when these rough graphene films are transferred to supporting substrates such as the holey carbon films of EM grids[4,11,24].

In view of this, we replaced the copper foil with Cu(111)/sapphire wafer as a growth substrate (Fig. 2d). The Cu(111) substrate can eliminate the rolling lines and inhibit the formation of step bunches and wrinkles[25,26], thus giving rise to the atom-scale flatness of graphene growth. The average surface roughness ($R_a$) of graphene on the wafer is only 0.28 nm, and no sharp step bunches or wrinkles are observed (Fig. 2e and Extended Data Fig. 3). The height difference of the step on the graphene wafer is decreased to ~1 nm, which is negligible compared with that of rough graphene on copper foil (~10–30 nm) (Fig. 2c).

To achieve the suspended UFG we transferred the UFG onto EM grids using a face-to-face transfer method (Methods). The ultraflat surface increased the contact area between the graphene and EM grids, and thus improved the interfacial contact (Supplementary Fig. 1), enabling the scalable direct transfer of wafer-scale graphene onto the EM grids with a high yield (Fig. 2f,g). The suspended UFG membranes on the grid had ultraclean and single-crystal surfaces with a high statistical intactness of ~98% (Supplementary Figs. 2 and 3). Importantly, the suspended graphene membranes remained ultraflat, with almost no noticeable wrinkles on the uniform graphene surfaces (Fig. 2g). The height variation of the suspended UFG was down to ~2 nm, which is significantly smaller than that of suspended rough graphene (~10–20 nm) transferred from copper foil (Fig. 2g,h and Extended Data Fig. 4), indicating that the flatness of UFG grown on Cu(111)/sapphire wafer can be well maintained after being transferred onto EM grids.

### Pre-tensioned ultraflat graphene enables uniform ice

The single-crystal suspended UFG bridges the gap between chemical vapor deposition graphene and ideal graphene, which has excellent mechanical properties. We measured the mechanical performance of the suspended graphene on grids using atomic force microscopy nanoindentation (Fig. 3a). The mechanical strength ($\sigma$) and Young's modulus ($E$) of UFG were 145 ± 13 GPa and 933 ± 171 GPa, respectively, which are comparable to the values of near-ideal graphene exfoliated from graphite ($\sigma$ = 130 ± 10 GPa, $E$ = 1010 ± 15 GPa) and much higher than those of rough graphene (Fig. 3b, Extended Data Fig. 5 and Supplementary Table 1). From the force–displacement curves we found that UFG had better resistance to deformation than rough graphene

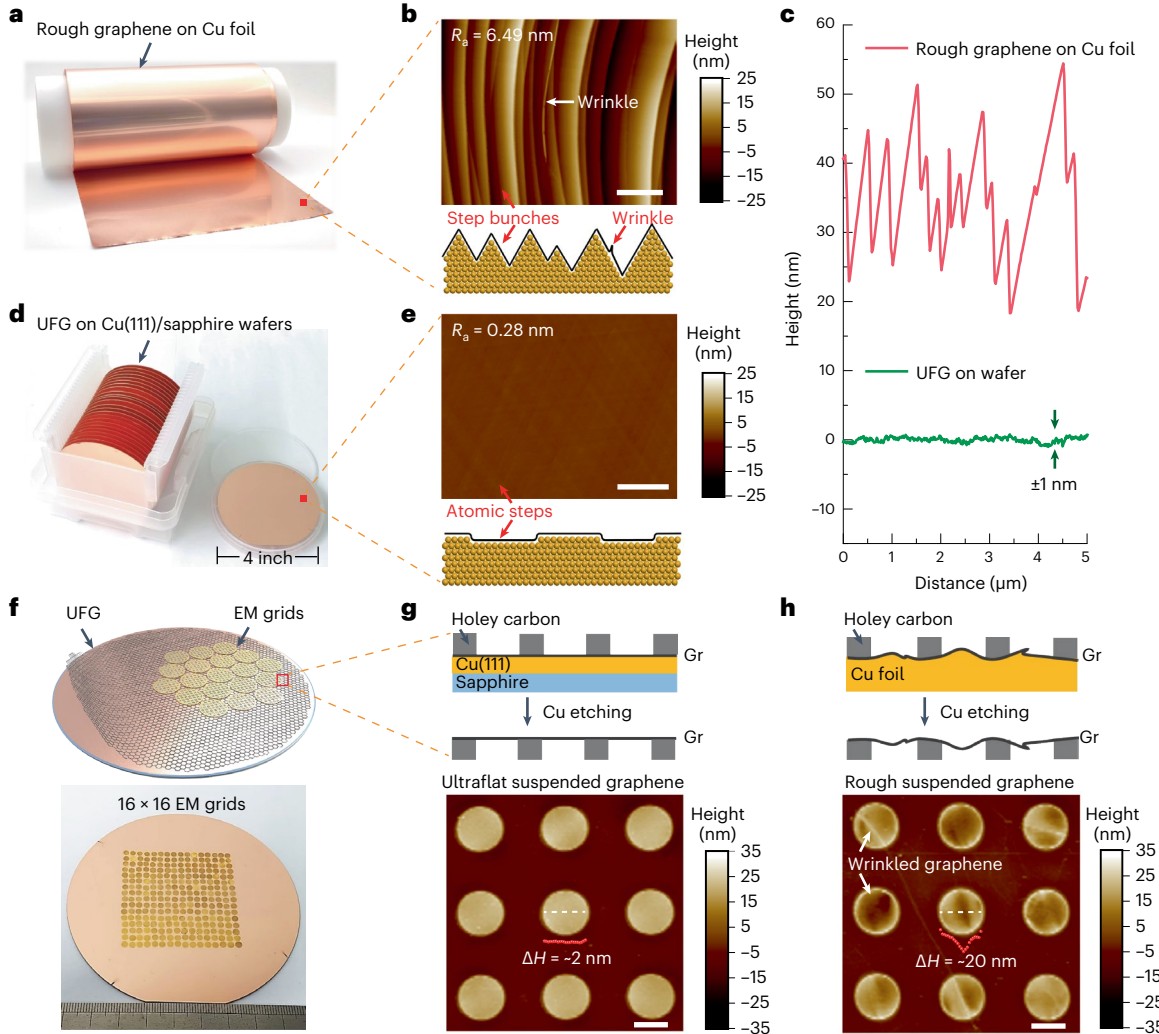

**Fig. 2 | Design and scalable preparation of ultraflat suspended graphene membranes. a**, Photograph of graphene (Gr) film grown on the copper foil. **b**, Typical AFM image (top) and corresponding schematic diagram (bottom) showing the rough graphene surface with dense step bunches and wrinkles on the copper foil. **c**, Typical smoothness of graphene films grown on the copper foil and Cu(111)/sapphire wafer. **d**, Photograph of UFG grown on the Cu(111)/sapphire wafers. **e**, Typical AFM image (top) and corresponding schematic diagram (bottom) showing the atomically flat surface of graphene on the wafer.

**f**, Schematic illustration (top) and photograph (bottom) showing the face-to-face transfer of wafer-scale UFG film onto the EM grids. **g**, Schematic diagram (top) and typical AFM image (bottom) of ultraflat suspended graphene transferred from the Cu(111)/sapphire wafer. The height variation along the white dashed line was plotted as red spots, and Δ$H$ was the maximum height difference. **h**, Schematic diagram (top) and typical AFM image (bottom) of rough suspended graphene transferred from copper foil. Scale bars, 1 μm.

(Extended Data Fig. 5e), and the corresponding pre-tension values of suspended UFG and suspended rough graphene were ~0.2 N m⁻¹ and ~0 N m⁻¹, respectively (Fig. 3c). The pre-tension in suspended UFG lies mainly in the fact that the periphery of the graphene membrane is attracted by the sidewalls of the holes in holey film (insets of Extended Data Fig. 4g,h). The pre-tension induced by the interaction between the graphene membrane and the sidewalls of holey substrates has been reported in previous experiments[27,28] and molecular dynamics simulations[29]. The pre-tension value of 0.2 N m⁻¹ of UFG is higher than the mechanical strength of many conventional materials[28], and helps to restrain the out-of-plane deformation of suspended graphene upon mechanical perturbations.

The pre-tension of the graphene plays a critical role in the preparation of cryo-EM specimens with thin vitreous ice. In the preparation procedure, shear force is one of the key factors that influence the formation of thin liquid film[2,8,30]. The shear stress imposed on the film could be on the order of kilopascals (kPa) when the liquid film thickness approaches tens of nanometers for high-resolution cryo-EM[8]. Based on

this, we constructed a theoretical simulation to investigate the influence of shear force on the suspended graphene. For the suspended UFG with a pre-tension of 0.2 N m⁻¹, the suspended membranes remained ultraflat without any distortion under a shear stress of ~10–100 kPa (Fig. 3d and Supplementary Fig. 4). In contrast, the wrinkle amplitude of suspended rough graphene significantly increased under the same shear stress, and the height differences of suspended graphene could increase to ~10–30 nm with a stress of ~10–100 kPa (Fig. 3e and Supplementary Fig. 5).

Our experimental results corresponded well with the simulations. After the wicking of excess solution from the suspended graphene surface (that is, the blotting procedure during cryo-EM specimen preparation), a thin liquid film is formed and then vitrified as the vitreous ice layer. Under cryo-EM, the vitreous ice on pre-tensioned UFG had evenly distributed intensities across the holes, indicating a uniform ice layer without wrinkles (Fig. 3f and Extended Data Fig. 6a). In contrast, the vitreous ice on the rough graphene had a wavy morphology with inconsistent contrast in the cryo-EM image (Fig. 3g and Extended

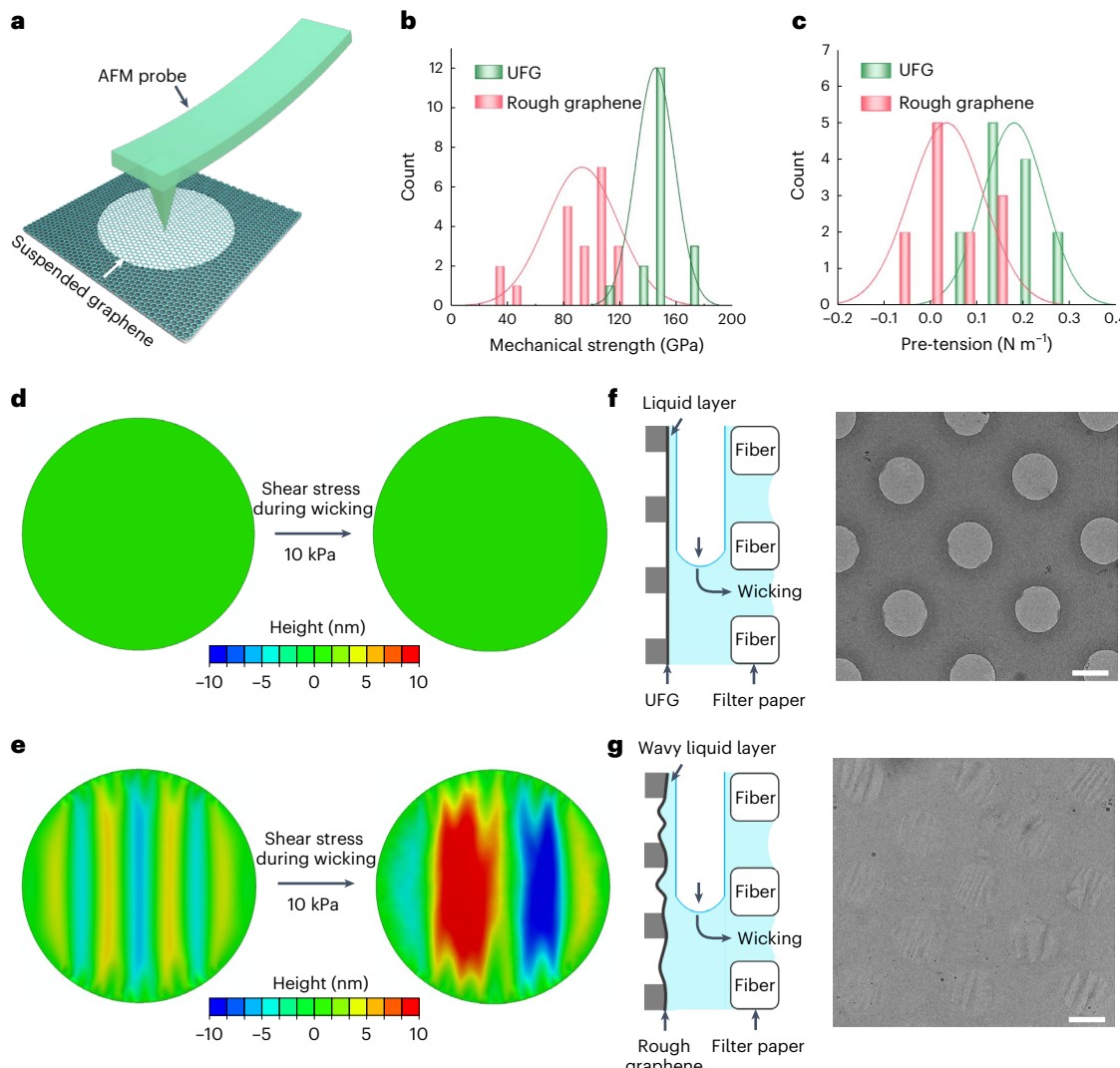

**Fig. 3 | Pre-tensioned ultraflat suspended graphene enables uniform ice.**
**a**, Schematic illustration of AFM nanoindentation. The AFM probe is used
to indent the center of the suspended graphene membrane. **b,c**, Statistical
histograms of mechanical strength (**b**) and pre-tension (**c**) for UFG and rough
graphene. **d,e**, Finite element simulations of UFG with a pre-tension of 0.2 N m⁻¹
(**d**) and rough graphene without pre-tension (**e**) under a shear stress of -10 kPa.

The UFG remained flat after being sheared, while the ripple amplitude of rough
graphene increased under the same shear stress. **f,g**, Left: schematic diagrams of
the wicking process, during which excess solution is wicked away, leaving behind
a thin liquid layer on the UFG (**f**) and rough graphene (**g**). Right: cryo-EM images
showing the uniform vitreous ice on the UFG (**f**) and wavy vitreous ice on the
rough graphene (**g**). Scale bars, 1 μm.

Data Fig. 6b). Given that the graphene was almost background free
in cryo-EM images, the difference in image contrast on the rough gra-
phene resulted mainly from the inconsistent ice thickness.

### Uniform thin ice improves image quality

To further characterize the ice behavior and particle distribution on
the UFG support, we used the UFG grids for cryo-EM specimen prepa-
ration of macromolecules, that is, the 20S proteasome, hemoglobin
and streptavidin. Here, we used rough graphene-supported specimens
as a control group. First, we imaged the cryo-specimen at different
tilt angles (Fig. 4a) to perform the cryo-ET analysis and found that at
high-angle tilt (+60° or −60°) the wrinkle features were clearly recorded
on the rough graphene-supported specimen, although these features
were negligible at 0° tilt (Fig. 4a). Using this tilt series we reconstructed
the tomogram of the rough graphene-supported specimen and plot-
ted the 20S proteasome particles in the ice (Fig. 4b). The ice of rough
graphene-supported specimens had a thickness variation of 5–20 nm,
generating varied background noise at the 100 nm lateral scale.

The protein particles were mainly adsorbed onto the graphene surface,
taking on a wave-like distribution (Fig. 4b). The non-coplanar particle
distribution (Fig. 4c) therefore resulted in different focus values of
individual molecules from the same micrograph without tilt, requir-
ing individual and careful refining during data processing. As well as
the cryo-ET analysis we also determined the ice thickness of the rough
graphene-supported specimen by filtering the inelastic scattering
electrons with an energy filter (Extended Data Fig. 7)[31]. Consistent with
the cryo-ET results, the ice was wave shaped with a thickness variation
of 5–20 nm (Fig. 4d), comparable to the aforementioned height varia-
tion of suspended rough graphene.

In contrast to rough graphene, the UFG-supported specimen had
uniform and flat ice, with no wrinkle-like features to overlap the target
particle signals at a high-angle tilt (Fig. 4e). The ice thickness retained
homogeneity, bounded by flat graphene and the air–water interface,
as shown on cryo-ET of the UFG-supported specimen (Fig. 4f). The 20S
proteasome particles were distributed at the same defocus level on
the flat graphene surface in the untilted specimen (Fig. 4f,g). The ice

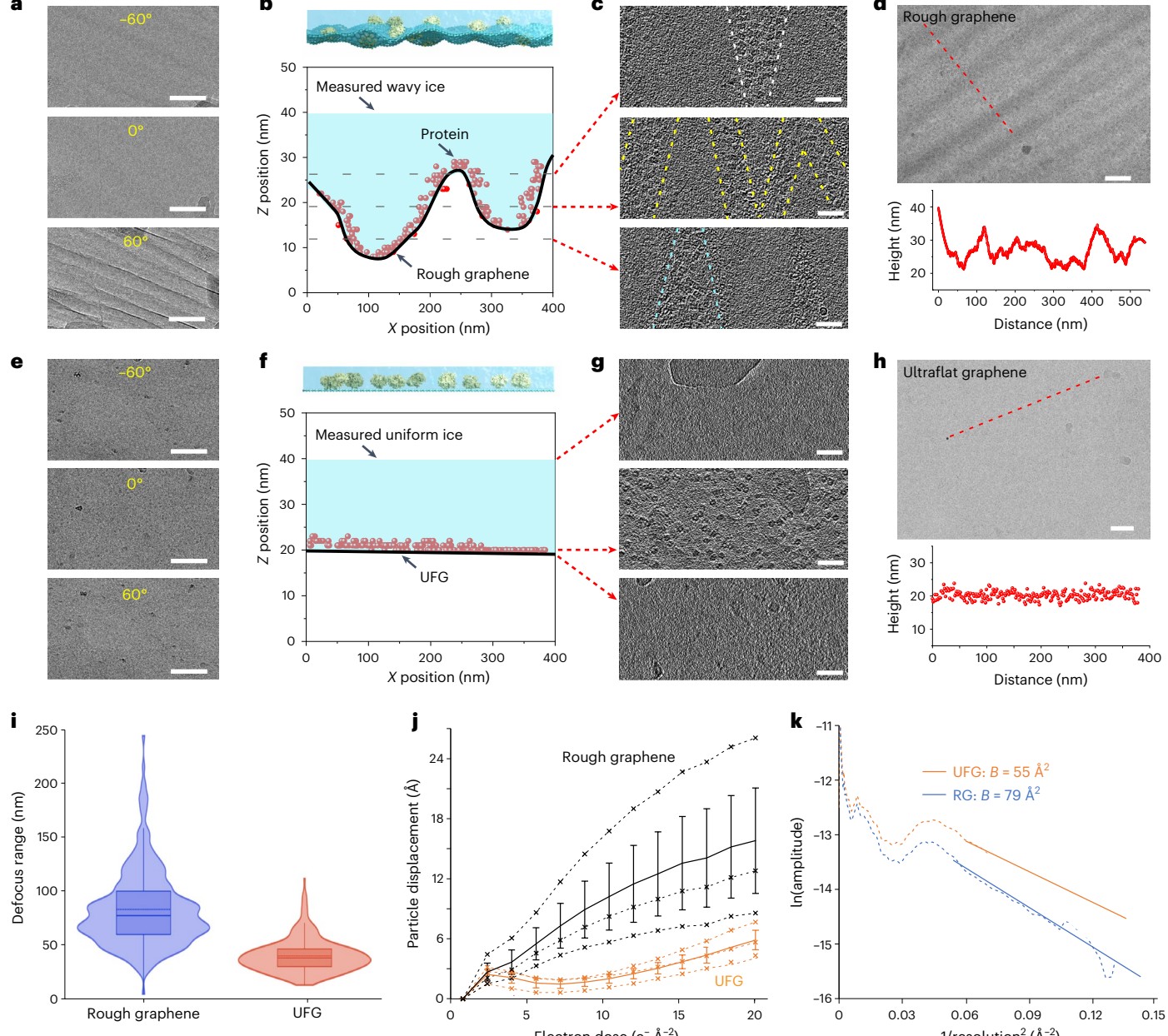

**Fig. 4 | Cryo-EM characterization of the vitrified ice and particle distribution on UFG and rough graphene. a**, Typical micrographs acquired at different tilt angles for a rough graphene-supported cryo-specimen. The wrinkles were obviously visible at high tilt angles and submerged the particles' signals. Scale bars, 100 nm. **b**, Top: schematic diagram of a rough graphene-supported specimen. Bottom: plot of 20S proteasome particle position in ice supported by suspended rough graphene, as per the cryo-ET reconstruction. Protein particles were mainly distributed on the graphene surface, with a height variation of ~20 nm. Every spot indicated one individual 20S proteasome particle. **c**, Three selected slices through the z axis from the tomogram of a rough graphene-supported cryo-specimen, the relative locations of which are indicated by the dotted lines in **b**. Scale bars, 50 nm. **d**, Top: typical micrograph of a rough graphene-supported cryo-specimen imaged with an energy filter. Bottom: ice thickness distribution across the dotted line in the upper micrograph. Scale bar, 100 nm. **e**, Representative micrographs acquired at different tilt angles for a UFG-supported cryo-specimen. No visible wrinkles were observed in the −60° or 60° tilted micrographs. Scale bars, 100 nm. **f**, Top: schematic diagram of the UFG-supported specimen. Bottom: protein particle position distribution on the UFG. The particles were adsorbed onto the graphene surface in the same plane. **g**, Three selected slices through the z axis from the tomogram of the UFG-

supported cryo-specimen, the relative locations of which are indicated by the red arrows in **f**. Scale bars, 50 nm. **h**, Top: typical micrograph of a UFG-supported cryo-specimen imaged with an energy filter. Bottom: ice thickness distribution across the dotted line in the upper micrograph. Scale bar, 50 nm. **i**, Defocus range distribution of micrographs acquired from rough graphene-supported (blue) and UFG-supported (orange) specimens. A total of 324 and 313 micrographs were collected and used here for rough graphene and UFG, respectively. The dotted lines in the boxes indicate the mean, and the solid lines indicate the median. The boundaries of the boxes are the first quartile (median of the lower half of the data) and the third quartile (median of the upper half), and the whiskers of the boxes define the upper and lower bounds. **j**, Particle motion with accumulated electron dose on rough graphene (black) and UFG (orange). Every dotted line was generated by averaging the displacement of thousands of particles. We performed three repeated measurements (dotted lines) for both rough graphene and UFG, using 2,001, 2,056 and 2,043 particles for rough graphene, and 1,910, 2,013 and 1,769 particles for UFG, respectively. The solid lines are the average of these dotted lines, and the error bars are the standard deviations. **k**, Guinier plots (dotted curves) assessing the image quality of rough graphene-supported (blue) and UFG-supported (orange) specimens. The solid lines are the linear fits at high resolution, and their slopes were used to calculate the B factor: $B = 4 \times$ slope.

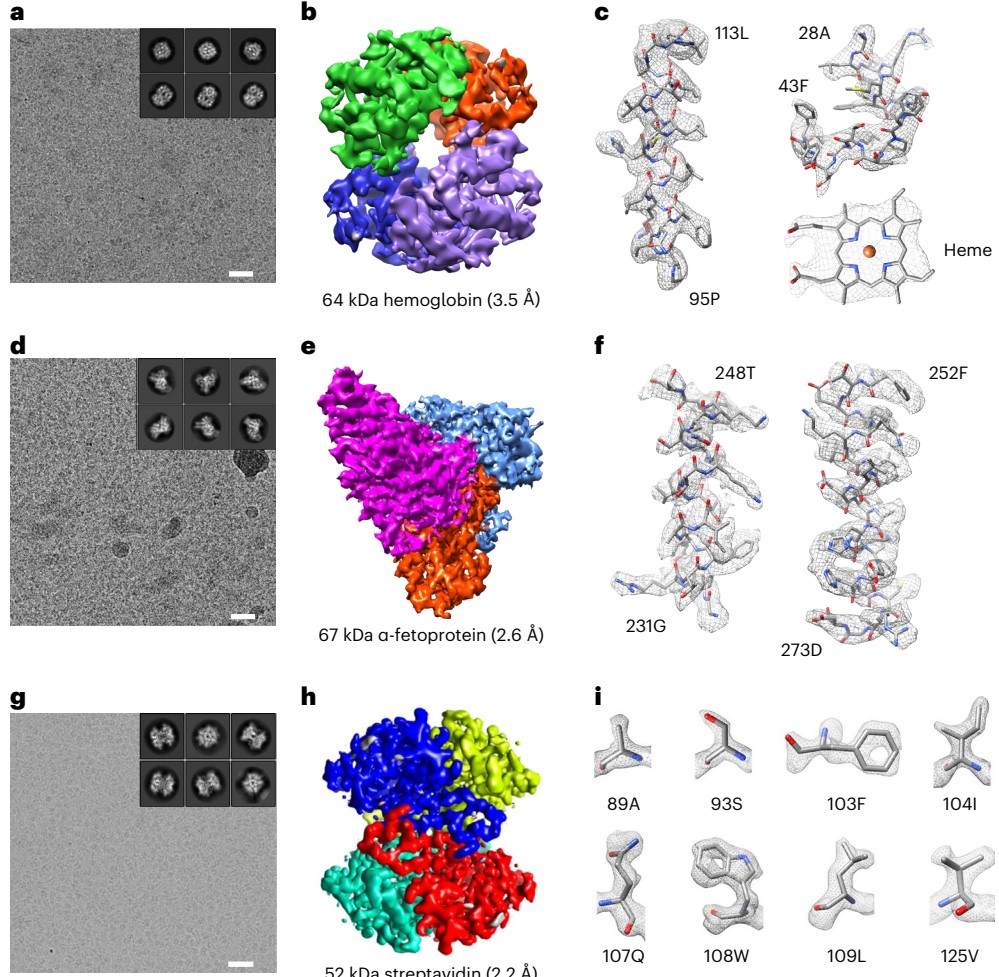

**Fig. 5 | High-resolution structural determination of small macromolecules by single-particle cryo-EM. a,d,g,** Typical cryo-EM micrographs of hemoglobin (**a**), α-fetoprotein (**d**) and streptavidin (**g**) supported by UFG. The insets show representative 2D class averages of the particle images Scale bars, 20 nm. **b,e,h,** The density maps of hemoglobin at 3.5 Å resolution, with individual monomers indicated by different colors (**b**), α-fetoprotein at 2.6 Å resolution, with individual domains indicated by different colors (**e**), and streptavidin at 2.2 Å resolution, with individual monomers indicated by different colors (**h**). **c,f,i,** Selected densities from the hemoglobin map (**c**) and α-fetoprotein map (**f**) with the corresponding atomic models docked; and the density of several amino acid residues extracted from the 2.2 Å streptavidin map with the corresponding atomic models of the residues (**i**).

thickness was estimated to be ~20 nm by calculating the space between the air–water interface (Fig. 4g, upper) and the graphene support (Fig. 4g, down), which was suitable for embedding protein particles without introducing extra noise. We also characterized the ice thickness variation using an energy filter and found that the ice thickness was consistently ~20 nm across the imaged region with much smaller fluctuation than that of the rough graphene-supported specimen (Fig. 4h).

Furthermore, we collected single-particle datasets of rough graphene- and UFG-supported 20S proteasomes (Supplementary Table 2), and calculated the defocus range of all micrographs based on the local defocus values of particles in individual micrographs. The defocus range of micrographs acquired from the rough graphene-supported specimen varied much more than that from the UFG-supported specimen, with a mean defocus range of 84.0 nm on rough graphene versus 40.5 nm on UFG (Fig. 4i). Moreover, the particle motion on the UFG induced by the electron beam was significantly smaller than that on rough graphene (Fig. 4j), owing to the ice layer on pre-tensioned UFG becoming more resistant to deformation during cryo-EM imaging in accordance with the physical theory of ice bending[32,33] (detailed discussions about the effect of pre-tension on the particle motion reduction are given in Methods). The uniformly thin ice with deformation resistance improved the image quality, as shown by the *B* factor extracted from the Guinier plots, which is defined as the natural logarithm of the structure factor amplitude against 1/resolution[2] and which describes the contrast loss with frequency. The extracted *B* factor reflects the decay rate of high-resolution information and signal-to-noise ratio. We found that the UFG-supported specimen had better high-resolution amplitude with a much smaller *B* factor of 55 Å² compared with the 79 Å² of rough graphene (Fig. 4k). Moreover, the calculated Henderson–Rosenthal *B* factor[34] on UFG was also smaller than that on rough graphene, enabling the higher-resolution structure determination with UFG when using exactly the same number of particles (Extended Data Fig. 8). These results collectively show that UFG enables the production of a uniformly thin ice and improves the image quality for cryo-EM analysis.

## Cryo-EM reconstruction of small macromolecules

Structural determination of biomolecules of small molecular weight (<100 kDa), due to the poor signal-to-noise ratio, remains a practical limit in cryo-EM[5,35]. To verify the imaging quality enabled by UFG support, we prepared cryo-EM specimens using 64 kDa hemoglobin, 67 kDa α-fetoprotein and 52 kDa streptavidin applied to the UFG and collected cryo-EM datasets for single-particle 3D reconstruction (Supplementary Table 2). All of these three samples of small molecular

weight were mono-dispersed on the UFG with high contrast (Fig. 5). The two-dimensional (2D) classification of hemoglobin particles showed a rich distribution of orientations (Fig. 5a, inset) that enabled a final reconstruction to be obtained at a resolution of 3.5 Å (Fig. 5b and Extended Data Fig. 9a,d). At this resolution the side chains and bound heme molecule could be clearly identified (Fig. 5c). The homologous structures of α-fetoprotein with no symmetry have previously been determined using X-ray crystallography[36,37]. Here, we collected 1 day cryo-EM datasets (2,290 micrographs) and followed the standard data-processing workflow to achieve a 2.6 Å resolution reconstruction (Fig. 5d–f and Extended Data Fig. 9b,e). In the cryo-EM reconstruction of streptavidin, fine details were obtained after a routine 2D classification process (Fig. 5g, inset). From the dataset we were able to obtain a 3D reconstruction of streptavidin at a resolution of 2.2 Å (Fig. 5h and Extended Data Figs. 9c,f,10). To our knowledge this is a high resolution achieved by cryo-EM for this size of protein[4,6,7,38–40]. Streptavidin is composed of four monomers, the densities of which were clearly resolved on the 2.2 Å map (Fig. 5h). Furthermore, we were able to identify the water molecules from the high-resolution EM map (Supplementary Fig. 6). All of the side chains of the amino acid residues were clearly assigned (Fig. 5i). For phenylalanine or tryptophan, the benzene at their side chains appeared as a ring-like density with an obvious hole in the center. Even for alanine, which has a minimal side chain, its methylene group was unambiguously visualized on the EM map (Fig. 5i). The 2.2 Å cryo-EM reconstruction of streptavidin indicates that the UFG support can improve cryo-EM imaging quality and that it has the potential to facilitate the determination of atomic-resolution reconstruction of small-molecular-weight biomolecules via standard data processing.

## Discussion

We prepared suspended UFG using graphene grown on a Cu(111)/sapphire wafer via the face-to-face transfer method with a high yield. The UFG had superior mechanical strength compared with the commonly used graphene-based cryo-EM supports. Importantly, we show that the flatness of the support film significantly influences the uniformity of ice thickness and the height distribution of particles in cryo-EM specimens. The resistance of UFG to deformation enables better control of uniformly thin ice and improves the image quality. We generated a cryo-EM map of 67 kDa α-fetoprotein with no symmetry at a resolution of 2.6 Å and achieved a resolution of 2.2 Å for 52 kDa streptavidin using such UFG grids.

Given that the uniform and thin ice on UFG eliminated the wavy features at high-angle tilt, UFG is a promising supporting substrate for cryo-ET or cryo-EM analysis with regard to particle preferential orientation[41], for which tilted datasets are typically required. Moreover, UFG can provide a flat and uniform interaction surface with functional ligands, thus enabling a more controllable bioactive functionalization for high-affinity and bio-friendly recognition of the target biomolecules. As well as its use in atomic-resolution EM imaging, the design of UFG could be generalized to other 2D materials to further extend the applications to drug discovery, high-performance electronic devices and separation membranes.

## Online content

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

## Methods

A step-by-step protocol is available in the Supplementary Information.

### Production of wafer-scale Cu(111)/sapphire substrate

The single-side polished C-plane (0001) sapphire wafers were purchased from Jiangsu Helios. The sapphire wafer had an ultraflat surface with an average roughness of <0.2 nm and a miscut angle of <0.5°. Before Cu film deposition, the sapphire wafers were annealed at 1,000 °C for 6 h in an oxygen atmosphere. Then, a 500-nm-thick Cu film was deposited onto the sapphire wafer by magnetron sputtering within 30 min using the physical vapor deposition method (Sputter film, SF2) with a radiofrequency power of ~300 W and a basal pressure of ~0.15 Pa. Subsequently, the wafers were annealed in a tube furnace at 1,000 °C with 500 standard cubic centimeters per minute (sccm) Ar and 10 sccm $H_2$ at atmospheric pressure for 1 h to seal the Cu(111) thin film to the sapphire substrate.

### Chemical vapor deposition growth of ultraflat graphene on Cu(111)/sapphire

The Cu(111)/sapphire wafer was placed on the surface of the other sapphire wafer (Supplementary Fig. 7) in a tube furnace. To synthesize the 4 inch UFG, the Cu(111) wafer was heated to 1,000 °C within 60 min under a flow of 2,000 sccm Ar and 40 sccm $H_2$ at atmospheric pressure, and then 40 sccm $CH_4$ (0.1% diluted in Ar) was introduced for graphene growth. Usually, it took 2 h for graphene to fully cover the surface of the Cu(111) wafer. $CH_4$ gas flow was shut down as soon as the growth was completed, and the sample was then cooled to room temperature.

### Chemical vapor deposition growth of graphene on copper foil

Commercially available Cu foil (Alfa Aesar 46986 or 46365) was used to grow graphene film. The Cu foil was first electrochemically polished in phosphoric acid and ethylene glycol solution (v/v 3:1) with a voltage of 2.4 V for 12 min. The polished copper foil was rinsed with deionized water and ethanol in sequence to remove the electrolyte. A nitrogen gun can be used to facilitate the drying process of polished copper foil. The copper foil was then annealed at 1,000 °C for 0.5 h in a tube furnace under a flow of 100 sccm $H_2$ with a corresponding pressure of ~100 Pa. Subsequently, 1 sccm $CH_4$ was introduced into the tube furnace for graphene growth. After 1 h the graphene film was synthesized on the copper foil. Finally, the graphene/copper foil was pulled out quickly from the heating area of the tube furnace and cooled down to room temperature under the flow of $H_2$.

### Transfer of graphene onto transmission electron microscopy grids

**Face-to-face transfer.** Commercial Au holey carbon grids (Quantifoil, Au 300 or 200 mesh R1.2/1.3) were used to prepare the graphene grids. Note that the grids with copper bars should be avoided because the copper bars can be etched by the $(NH_4)_2S_2O_8$ aqueous solution. To transfer UFG from the Cu(111)/sapphire wafer to the transmission electron microscopy (TEM) grids we placed a batch of TEM grids on the wafer. We then dropped a few droplets of isopropanol to cover the surface of these TEM grids. After the solution was volatilized, the TEM grids and UFG were combined (Supplementary Fig. 8). The same procedure was carried out to combine the TEM grids and rough graphene on the copper foil.

**Etching.** The grids/graphene/Cu(111)/sapphire composite was submerged in a 1 M $(NH_4)_2S_2O_8$ aqueous solution to etch the Cu(111) film away, to produce graphene grids with UFG covering the holey carbon film. For the etching of the copper foil, the grids/graphene/copper foil composite was floated on the surface of an $(NH_4)_2S_2O_8$ solution because the copper foil was light enough to be supported by the surface tension of the solution[42].

**Rinsing.** Graphene grids were submerged in deionized water to wash away the inorganic salts. They were then transferred into isopropanol for further rinsing.

**Drying.** The graphene grids were typically dried in the super-clean room to avoid extra contaminants after being removed from the isopropyl alcohol.

### Glow discharging of graphene grids

To render the hydrophilic graphene grids, graphene grids were glow-discharged for ~12 s using the 'low' setting of a plasma cleaner (Harrick, PDC-32G) after the chamber was evacuated for 2 min.

### Atomic force microscopy nanoindentation

The mechanical properties of suspended graphene membranes were measured in nanoindentation experiments using an Asylum Cypher ES system. A single-crystal diamond probe (ART D300, SCD Probes) with a tip radius of ~10 nm was used, and the cantilever stiffness was 30.85 N m$^{-1}$ as calibrated using the Sader method[43]. A constant deflection rate of 500 nm s$^{-1}$ was used in all of the tests.

The pre-stress and elastic modulus were extracted from the indentation force versus depth data, using the previous model by Lee et al.[44]:

$$F = (\pi\sigma^{2D})\delta + \left(E^{2D}\frac{q^3}{r^2}\right)\delta^3,$$

where $F$ is the applied load, $\sigma^{2D}$ and $E^{2D}$ are the 2D pre-stress and Young's modulus of the nanosheets, respectively, $\delta$ is the indentation depth and $r$ is the radius of the microwells. The dimensionless constant $q = 1/(1.05 - 0.15v - 0.16v^2)$, in which $v = 0.165$ is the Poisson ratio for graphene[44]. The breaking force and mechanical strength of suspended graphene were directly measured on the graphene grids (Quantifoil R1.2/1.3). Young's modulus of suspended graphene and the pre-tension were directly measured on the rigid $Si_3N_4$ graphene grids (R1.2/1.3) to prevent the deformation of holey substrates during the nanoindentation.

### Modeling and simulations of graphene rippling

Finite element simulations were performed in ABAQUS software. A circular monolayer graphene sheet with a diameter of 1.2 µm was modeled as a linear elastic isotropic plate using the four-node shell (S4R) element. The following parameters were adopted: Young's modulus $E = 1,000$ GPa, $v = 0.165$ and thickness $t = 0.335$ nm. Two types of graphene models were prepared for comparison. For a UFG sheet, radial displacement is imposed on the circular boundary to impose a pre-stress $\sigma^{2D} = 0.2$ N m$^{-1}$. A rough graphene sheet is formed by squeezing the circular plate along $x$ (displacement equal to 10 nm), which leads to buckling of the plate with a ripple amplitude of ~5 nm. Both graphene sheets then underwent uniform in-plane shear of 10–100 kPa in the $x$ direction.

### Effect of pre-tension in UFG on the particle motion reduction

As noted by Naydenova et al.[32], the compressive stress will build up in the thin ice owing to the inhibited volume change when the liquid water cools rapidly and turns into vitreous ice. Naydenova et al. have shown that the largest volumetric strain in ice is $\left(\frac{\Delta V}{V}\right)_{max} \approx 0.06$. Assuming that ice is an isotropic linear elastic material, the radial compressive strain is about $\varepsilon_r = -\frac{1}{3}\left(\frac{\Delta V}{V}\right)_{max} \approx -0.02$. Taking $E_{ice} \approx 1$ GPa and $v_{ice} \approx 0.3$ as Young's modulus and Poisson's ratio of amorphous ice, respectively, there is a radial compressive stress of $\sigma_r = \frac{E_{ice}}{1-v_{ice}}\varepsilon_r \approx -0.029$ GPa in the ice layer (or equivalent to $N_r \approx -0.58$ N m$^{-1}$ in 2D stress form given that the ice layer thickness is 20 nm). Note that the negative sign means a compressive stress and strain.

According to the physical theory of ice bending[32], the critical compression for a clamped circular membrane to buckle can be approximated by $N_c = -\frac{14.682 E_c h_c^3}{12 r^2 (1 - v_c^2)}$ in which $E_c$, $h_c$ and $v_c$ are the effective Young's modulus, total thickness and effective Poisson's ratio of the UFG–ice composite layer, respectively. Applying the rule of mixtures, we can estimate $E_c \approx 16.4$ GPa and $v_c \approx 0.3$. Therefore, the critical compression for the UFG–ice layer to buckle is $N_c \approx -0.13$ N m$^{-1}$.

For UFG, our nanoindentation tests have shown an average pre-tension of approximately 0.2 N m$^{-1}$. Thus, the maximum effective compressive stress in the UFG–ice composite layer is $-0.58 + 0.2 = -0.38$ N m$^{-1}$. Therefore, the calculations above suggest that although the UFG–ice layer might still buckle eventually, the pre-tension in UFG can effectively delay the buckling, and the maximum compression in ice is also notably reduced (from $-0.58$ N m$^{-1}$ down to $-0.38$ N m$^{-1}$). In contrast, the rough graphene has no pre-tension to compensate for the compression in the ice layer, therefore the rough graphene–ice composite layer has much less resistance to buckling.

## AFM, SEM and STEM characterizations

The morphologies of graphene films on growth substrates and suspended graphene membranes on EM grids were characterized by atomic force microscopy (AFM) in tapping mode (Bruker Dimension Icon with Nanoscope V controller). The coverage of suspended graphene membranes on the graphene grid was collected using scanning electron microscopy (SEM) (2 kV, Hitachi S-4800), and the aberration-corrected scanning transmission electron microscopy (STEM) images of graphene were carried out using a Nion U-HERMS200 microscope at 60 kV.

## Cryo-EM specimen preparation

Streptavidin was purchased from New England Biolabs (catalog no. N7021S) and hemoglobin was purchased from Sigma-Aldrich (catalog no. H7379). For 20S proteasome purification[13] we first constructed a plasmid containing the α and β subunits of the 20S proteasome, and the amino terminus of the β subunit was His-tagged. The plasmid was then transformed into *Escherichia coli*-competent cells and left overnight for expression. The cells were centrifuged and sonicated to obtain the cell lysates, which were finally loaded onto a nickel column (GE Healthcare) for affinity purification. For α-fetoprotein we first transfected the recombinant vector of pCAG-*afp*-His-Strep into HEK-293F cells. After transfection the cells were cultured for 2 days and then collected by centrifugation. The pellets were resuspended in lysis buffer and loaded onto Strep-Tactin XT for affinity purification. We then performed gel filtration to obtain the purified α-fetoprotein sample[45].

A 3 μl drop of a sample solution containing 0.2 μM 20S proteasome or 0.1 μM streptavidin or 6 mg ml$^{-1}$ hemoglobin or 1 μM human α-fetoprotein was pipetted onto graphene grids, which had been previously glow-discharged. The grids were transferred into an FEI Vitrobot with humidity at 100% and temperature at 8 °C and blotted for 1 s with a force of −2. Afterward, the grid was plunge-frozen into liquid ethane and kept in liquid nitrogen for further cryo-EM imaging.

## Cryo-EM data collection and processing

We used AutoEMation Software[46] written by J. Lei at Tsinghua University to automatically collect single-particle datasets on an FEI Titan Krios (300 kV), equipped with an energy filter and a Gatan K3 summit detector, with an accumulated dose of 50 e$^-$ Å$^{-2}$. All of the movies contained 32 frames, which were motion-corrected using the MotionCor2 algorithm[47]. The contrast transfer function (CTF) values were estimated using CTFFIND4[48]. We then used Relion3.1[49] to select the particles and performed iterative 2D classification to discard bad particles. The remaining good particles were imported for 3D classification and refinement. For hemoglobin, C2 symmetry was imposed in the 3D refinement with a final particle number of 105,000 (Supplementary Table 2), and the reported resolution was 3.5 Å. To generate the atomic model of hemoglobin, the previously published coordinate (Protein Data Bank: 7VDE)[50] was docked into the cryo-EM map and real-space refined in PHENIX[51]. For α-fetoprotein we collected 2,290 micrographs and used 354,264 particles in the final 3D reconstruction to obtain a 2.6 Å density map. For streptavidin, D2 symmetry was applied in the 3D refinement step. The resolution (2.2 Å) was determined using Fourier shell correlation (FSC) = 0.143 as the cut-off criterion, and the final particle number used here is 260,390 (Supplementary Table 2). The Figures showing structural detail were generated in UCSF Chimera[52].

We performed CTF refinement in Relion3.0 to estimate the local defocus of particles in individual micrographs and plotted the defocus range. To measure the particle motion on graphene support, we performed Bayesian polishing to determine the particle coordinates in each frame. These coordinates were then used to calculate the root mean squared displacements of each particle and plotted versus dose. We carried out three particle-motion measurements of both UFG and rough graphene, each of which was based on thousands of particles.

## Cryo-electron tomography analysis

Cryo-ET tilt series were acquired using SerialEM software[53] from +60° to −60° with a step of 3°, on an FEI Titan Krios microscope (300 kV) equipped with a Gatan K2 camera. At each tilt angle, we collected movies containing 8 frames, with a sum dose of ~3 e$^-$ Å$^{-2}$ s$^{-1}$, and the total dose for every tilt series (+60° to −60°) was 120 e$^-$ Å$^{-2}$. The pixel size was 1.25 Å. The movies were first motion-corrected by MotionCor2[47] and then imported into Etomo[54] for alignment and reconstruction. The position of protein particles inside the ice was manually identified and plotted[13].

## Statistics and reproducibility

The experiments in Fig. 2b,e,g,h, Extended Data Figs. 1a,b,d,e, 2a,b,d,3a,d,4a–f,5a,b and Supplementary Figs. 1b–f, 2a–i,3a–c were repeated more than five times independently with similar results, including the growth of UFG and rough graphene, preparation and characterizations of suspended UFG and suspended rough graphene. The experiments related to Fig. 3f and Extended Data Fig. 6a for UFG and those related to Fig. 3g and Extended Data Fig. 6b for rough graphene were both repeated in more than five grids independently, with similar results. Fifty micrograph pairs for rough graphene (related to Fig. 4d and Extended Data Fig. 7) and 32 micrograph pairs for UFG (related to Fig. 4h), and five tilt-series datasets for rough graphene (related to Fig. 4a) and 10 tilt-series datasets for UFG (related to Fig. 4e) were collected, demonstrating similar results. For single-particle cryo-EM dataset collection, we collected 5,604 micrographs for hemoglobin (Fig. 5a), 2,290 micrographs for α-fetoprotein (Fig. 5d) and 2,332 micrographs for streptavidin (Fig. 5g).

## Reporting summary

Further information on research design is available in the Nature Portfolio Reporting Summary linked to this article.

## Data availability

The coordinates and density maps of streptavidin, hemoglobin and α-fetoprotein have been deposited in the RCSB Protein Data Bank (PDB) under the accession numbers 8GVK, 7XGY and 7YIM, and the EMDB under the accession numbers EMD-32099, EMD-33189 and EMD-33861, respectively. The raw cryo-EM dataset of streptavidin, containing the unaligned movies and particles, has been deposited into EMPIAR under accession number EMPIAR-11217. Any additional data supporting the findings in this manuscript are available from the corresponding authors upon reasonable request. Source data are provided with this paper.

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

## Acknowledgements

We thank J. Lei, X. Li, F. Yang and T. Liu at the Cryo-EM and High-Performance Computation platforms of Tsinghua University Branch of the National Protein Science Facility for technical support in cryo-EM data collection. N.L. thanks the Shuimu Tsinghua Scholar Program and Advanced Innovation Fellowship from Beijing Advanced Innovation Center for Structural Biology for support. L.Z. thanks Y. Hou, Z. Zhang and Z. Liu for fruitful discussions. We also thank the Electron Microscopy Laboratory and Molecular Materials and Nanofabrication Laboratory(MMNL) in the College of Chemistry at Peking University for the use of instruments. The work is supported by the National Natural Science Foundation of China (52021006 and T2188101 to H.L.P., 31825009 to H.-W.W., and 12172005, 11890681 and 11988102 to X.W.) and Beijing National Laboratory for Molecular Sciences (BNLMS-CXTD-202001 to H.L.P.). This work is also supported by grants 2016YFA0501100 and 2017YFA0503501 (to H.-W.W.) from the Ministry of Science and Technology of China, Z161100000116034 (to H.-W.W.) from the Beijing Municipal Science & Technology Commission and THE XPLORER PRIZE (to H.-W.W. and H.L.P.). N.L. was supported by the China Postdoctoral Science Foundation (2021M701919). The funders had no role in study design, data collection and analysis, decision to publish or preparation of the manuscript.

## Author contributions

L.Z., N.L., H.-W.W. and H.L.P. conceived the project and designed the experiments. L.Z. and X.Y.G. prepared the suspended ultraflat graphene membranes. N.L., K.L., C.W., Z.L., M.L., J.X., Y.L. and H.-W.W. carried out the cryo-EM characterizations and analysis. W.Z., L.Z. and X.W. performed the AFM nanoindentation tests, finite element simulation and mechanical analyses. R.Y. and B.D. synthesized the ultraflat graphene film. L.Z., X.Y.G., J.Z., X.G. and Y.Y contributed to the characterizations of suspended graphene. H.L.P., L.Z., N.L. and H.-W.W. wrote the manuscript. All authors discussed the results and edited the manuscript. All authors approved the final version of the manuscript.

## Competing interests

The authors declare no competing interests.

## Additional information

**Extended data** are available for this paper at https://doi.org/10.1038/s41592-022-01693-y.

**Correspondence and requests for materials** should be addressed to Nan Liu, Xiaoding Wei, Hong-Wei Wang or Hailin Peng.

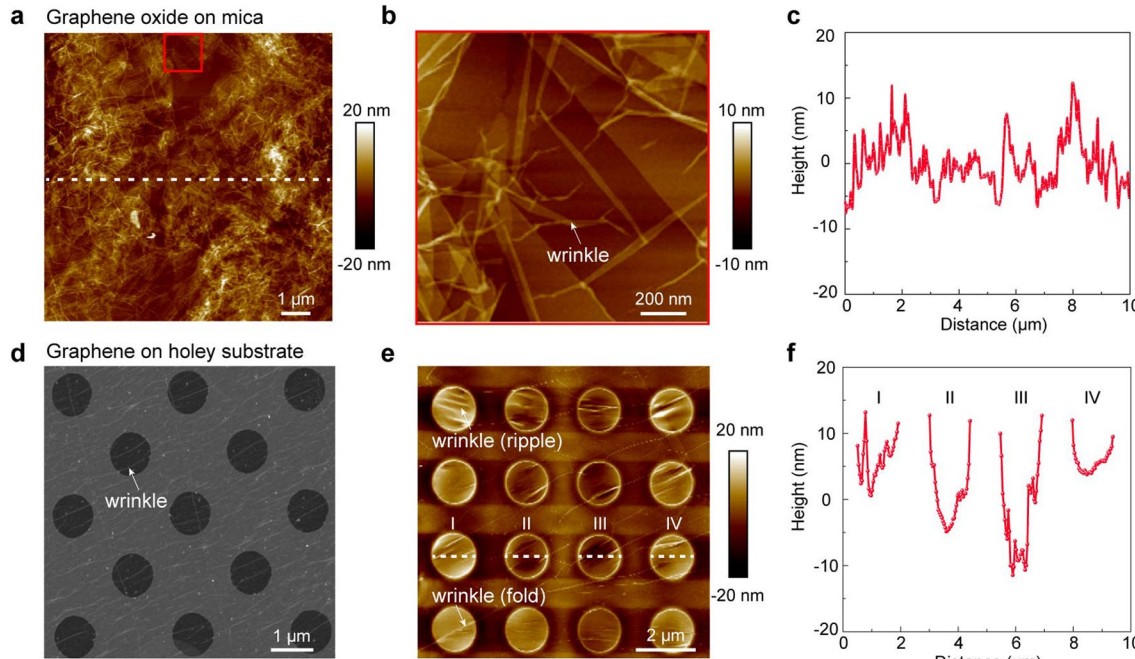

**Extended Data Fig. 1 | Roughness of graphene oxide on mica and suspended graphene membranes on the holey substrate. a**, Typical atomic force microscopy (AFM) image of graphene oxide nanosheets on mica. **b**, AFM image showing the thin area of graphene oxide from the marked region in (**a**). The arrow reveals the wrinkle of graphene oxide. **c**, Height profile along the white line as marked in (**a**). The maximum height difference of graphene oxide nanosheets can reach ~20 nm. **d**, Typical scanning electron microscopy (SEM) image of rough suspended graphene on a TEM grid, which was transferred from the CVD graphene grown on the copper foil. The arrow reveals wrinkles of rough graphene. **e**, Typical AFM image of rough suspended graphene on a TEM grid. The arrows reveal the wrinkles on the suspended graphene surface. **f**, Height profiles along the white lines as marked in (**e**). The maximum height difference of rough suspended graphene can be up to 10-20 nm. Source data are provided as a Source Data file.

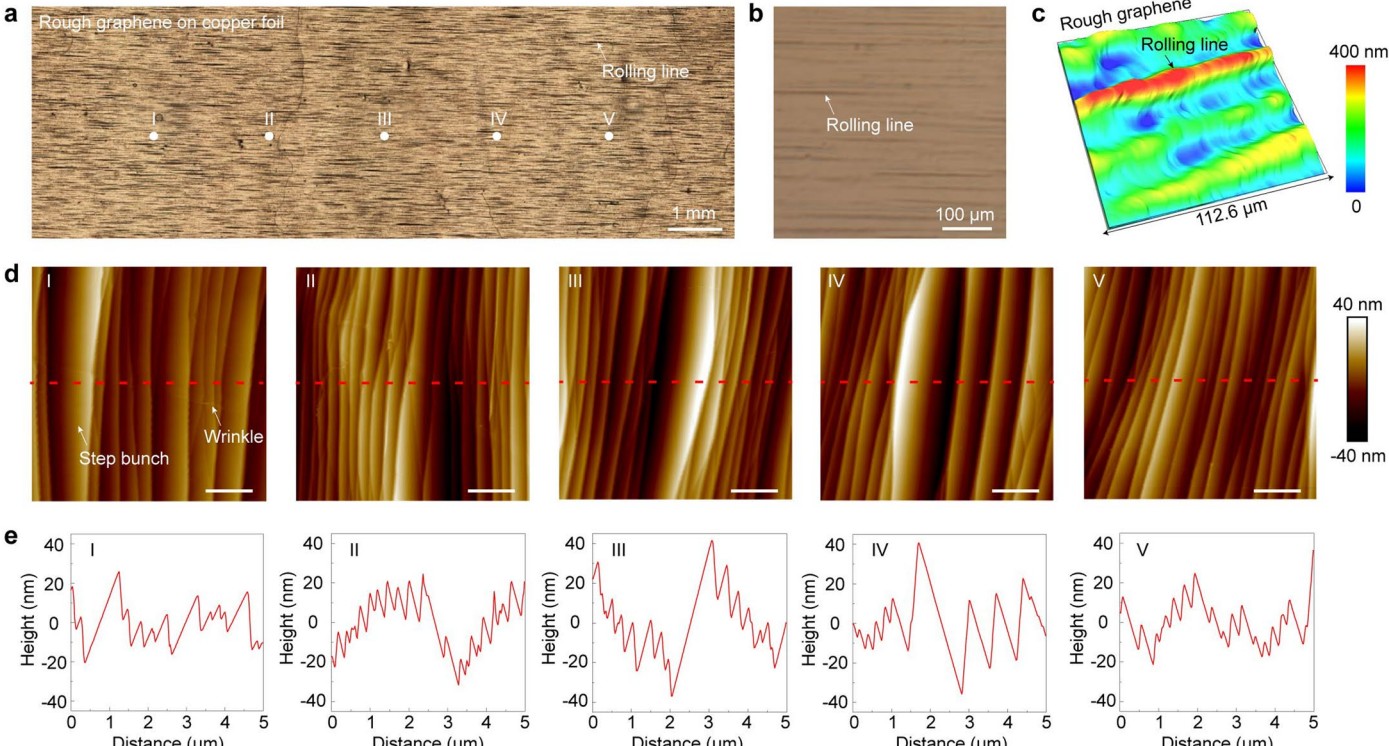

**Extended Data Fig. 2 | Rough graphene film on the copper foil. a**, A large-area optical microscopy (OM) image of rough graphene film on the copper foil. The arrow reveals a typical rolling line. **b**, A high-magnification OM image showing the rolling lines on the copper foil. **c**, White light interference (WLI) showing the height variations of rough graphene on the copper foil. The arrow reveals a rolling line where the height difference is ~400 nm. **d**, Typical AFM images revealing dense step bunches and wrinkles on the graphene surface from the marked regions in (**a**). **e**, Height profiles along the red lines in (**d**), demonstrating the height differences of step bunches can reach several tens of nanometers at a lateral scale of 5 μm. Scale bars, 1 μm. Source data are provided as a Source Data file.

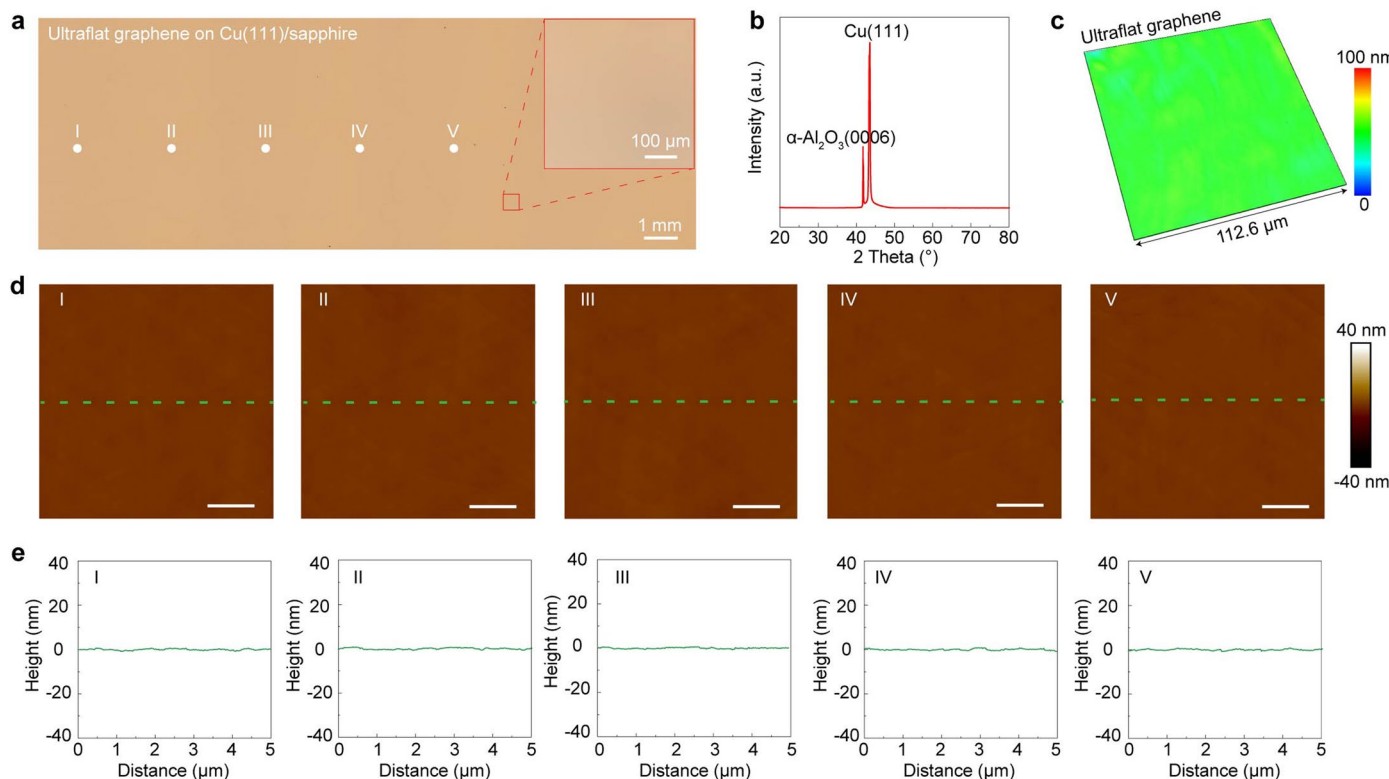

**Extended Data Fig. 3 | Ultraflat graphene on the Cu(111)/sapphire wafer.**
**a**, A large-area OM image of UFG on the wafer-scale Cu(111)/sapphire. Inset: the high-magnification OM image shows that no rolling lines were observed on the Cu(111)/sapphire. **b**, XRD pattern of Cu(111)/sapphire, proving the single-crystal Cu(111) film on the sapphire(α-Al$_2$O$_3$). **c**, White light interference (WLI) showing the ultraflat surface of graphene film on the Cu(111)/sapphire. **d**, Typical AFM images revealing the atomically flat graphene surface from the marked regions in (**a**). No step bunches and wrinkles were observed. **e**, Height profiles along the green lines in (**d**), showing that the maximum height variations of UFG are ±1 nm at a lateral scale of 5 μm. Scale bars, 1 μm. Source data are provided as a Source Data file.

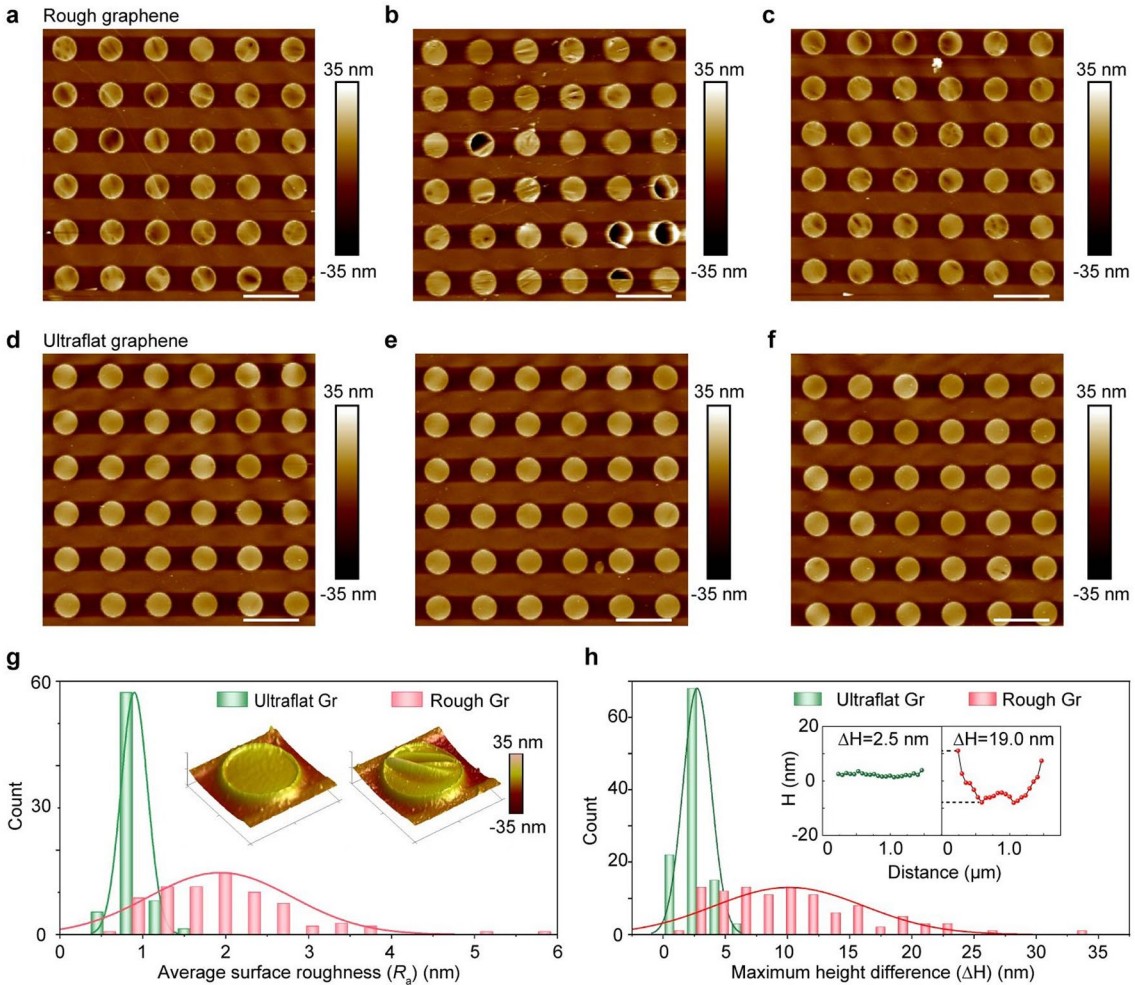

**Extended Data Fig. 4 | Surface roughness of UFG and rough graphene.**
**a-c**, AFM images of rough suspended graphene membranes, where dense
wrinkles were observed. The dark holes in (**b**) indicated the broken graphene
regions. Scale bars, 3 μm. **d-f**, AFM images showing the ultraflat suspended
graphene membranes with no wrinkles and superior coverage rate. Scale bars,
3 μm. **g**, Distributions of average surface roughness ($R_a$) of suspended UFG
membranes (olive) and rough graphene membranes (red). Insets: typical 3D AFM
images of suspended UFG (left) and rough graphene (right), respectively. The $R_a$
of ultraflat suspended graphene is -0.7 nm, apparently smaller than that of rough
suspended graphene (-2 nm). **h**, Distribution of maximum height difference (ΔH)
along the diameter of suspended UFG (olive) and rough graphene (red). Insets:
typical height profiles of suspended UFG and rough graphene, respectively. Most
ΔH of UFG distributed around 2 nm, while ΔH of rough graphene exhibited an
uneven distribution (10 ± 6 nm). Source data are provided as a Source Data file.

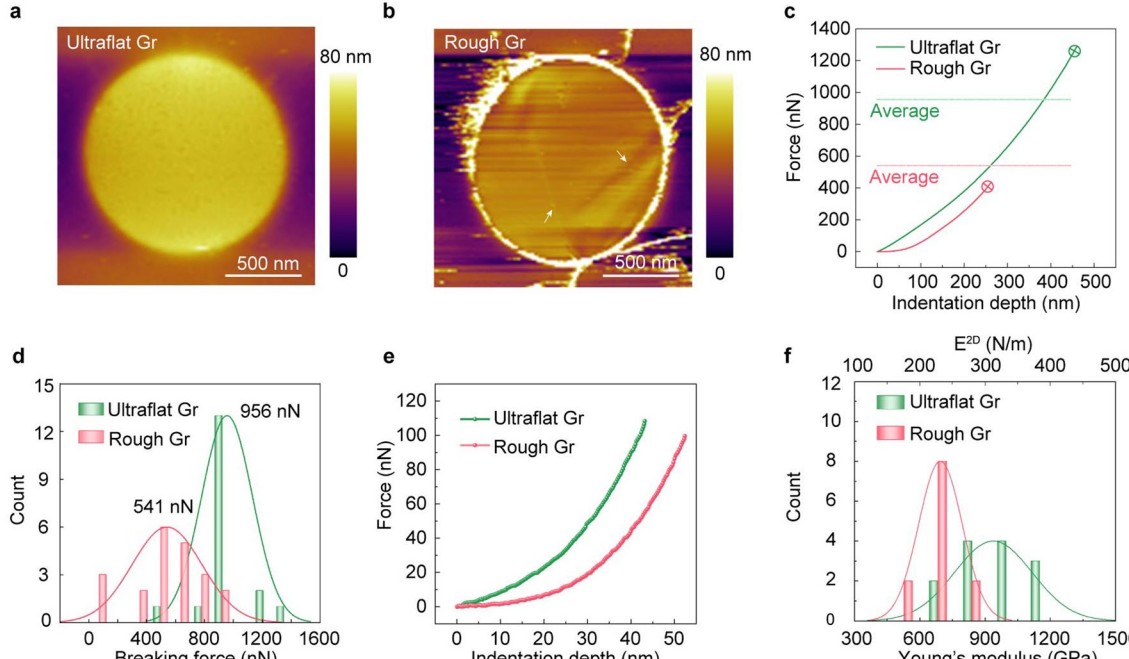

**Extended Data Fig. 5 | Comparison of mechanical properties between suspended UFG and rough graphene. a**, Typical AFM image of suspended UFG on the grid. **b**, Representative AFM image of suspended rough graphene. The arrows reveal wrinkles in rough graphene. **c**, Typical force-indentation depth curves of suspended UFG (olive) and rough graphene (red). Meanwhile, the average breaking forces of UFG and rough graphene were shown in the plot. **d**, Distribution of breaking forces of suspended UFG (olive) and rough graphene

(red). The average breaking force of UFG (~956 nN) was almost twice as large as that of rough graphene (~541 nN). **e**, Typical force-indentation depth curves showing the deformations of suspended UFG (olive) and rough graphene (red) at the initial stage of AFM indentation. **f**, Comparison of Young's modulus between UFG (olive) and rough graphene (red). Source data are provided as a Source Data file.

**a** Vitreous ice on ultraflat graphene

**b** Vitreous ice on rough graphene

Wavy ice

**Extended Data Fig. 6 | Square atlases of vitreous ice on the UFG and rough graphene. a**, Cryo-EM images revealing the uniform vitreous ice on the suspended UFG. The intensities in all these holes were evenly distributed. **b**, Cryo-EM images showing that the intensities cross individual holes were unevenly distributed, indicating non-uniform ice thickness. Scale bars, 5 μm.

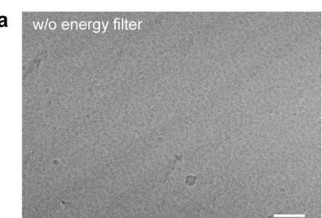
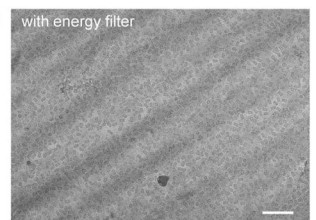

**Extended Data Fig. 7 | Cryo-EM characterizations of vitreous ice on rough graphene. a-b**, Cryo-EM images of vitreous ice on rough graphene without (**a**) and with (**b**) energy filter. Scale bars, 100 nm. The energy filter can significantly enhance the single-to-noise ratio by filtering the inelastic scattering electrons and has been used for determining the ice thickness via the intensity comparison of the filtered and non-filtered micrographs for the desired region. We imaged the rough graphene-supported specimen with an energy filter and found that the background intensity was obviously different across the imaging region (**b**). We further took another image at the same site without energy filter (**a**) and measured the intensities ratio of the non-energy-filtered and energy-filtered region, and finally obtained the ice thickness variation across the micrograph.

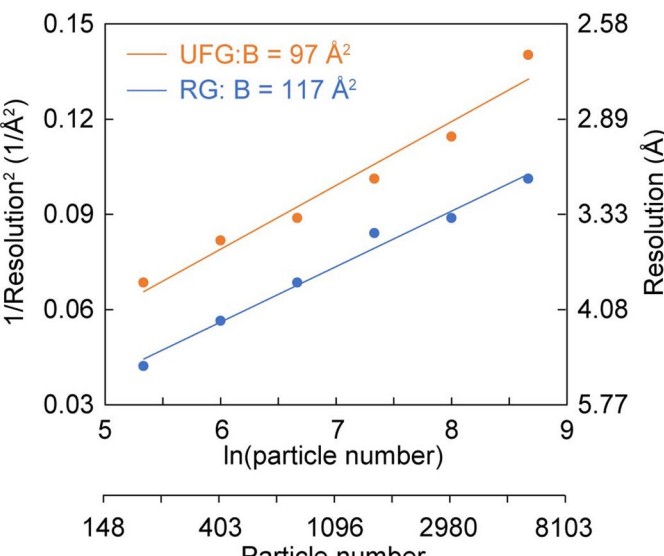

**Extended Data Fig. 8 | Performance comparison of UFG and rough graphene for cryo-EM imaging.** Performance as revealed by the number of 20S proteasome particles required for cryo-EM structure determination to achieve a certain resolution. The UFG (orange) exhibited superior performance in achieving a higher resolution with the same number of particles when compared with rough graphene (RG, blue). The Henderson $B$ factors were extracted from the slopes of fitted lines ($B$ factor=2/slope). The $B$ factor of particles on UFG (97) was smaller than that on rough graphene (117). Source data are provided as a Source Data file.

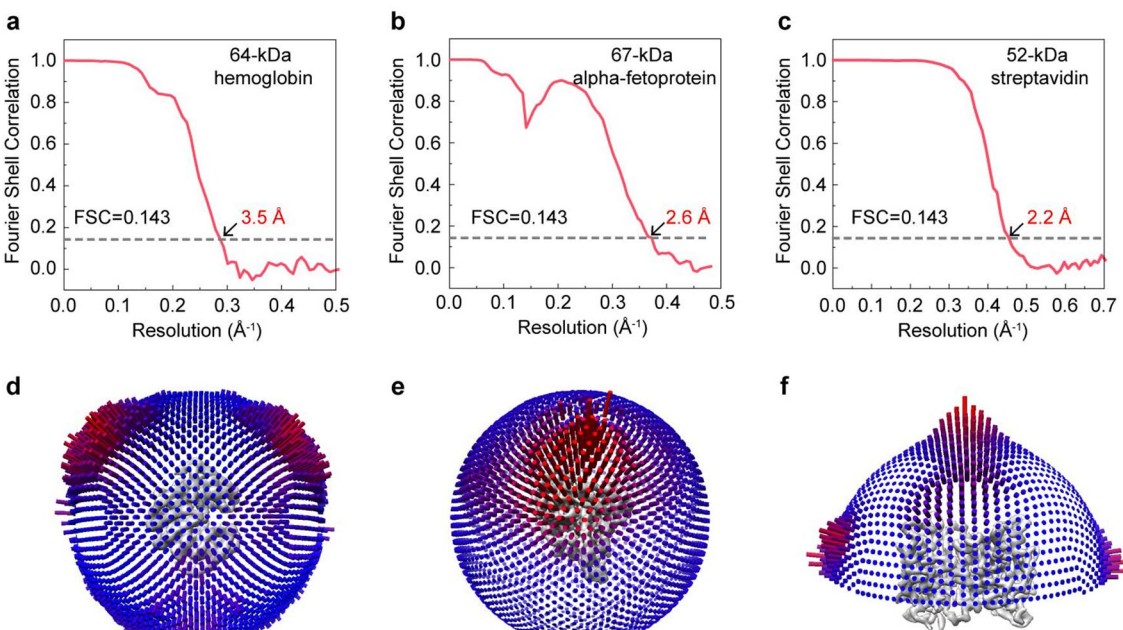

**Extended Data Fig. 9 | Resolution and particle orientational distribution of biomolecules with small molecular weight (<100 kDa). a-c**, The Gold-Standard Fourier Shell Correlation (FSC) curves of 64-kDa hemoglobin (**a**), 67-kDa alpha-fetoprotein (**b**), and 52-kDa streptavidin (**c**) reconstructions, respectively. FSC = 0.143 was indicated by the dotted line, used for estimating the resolution. **d-f**, Particle orientational distributions of hemoglobin (**d**), alpha-fetoprotein (**e**) and streptavidin (**f**), respectively. Source data are provided as a Source Data file.

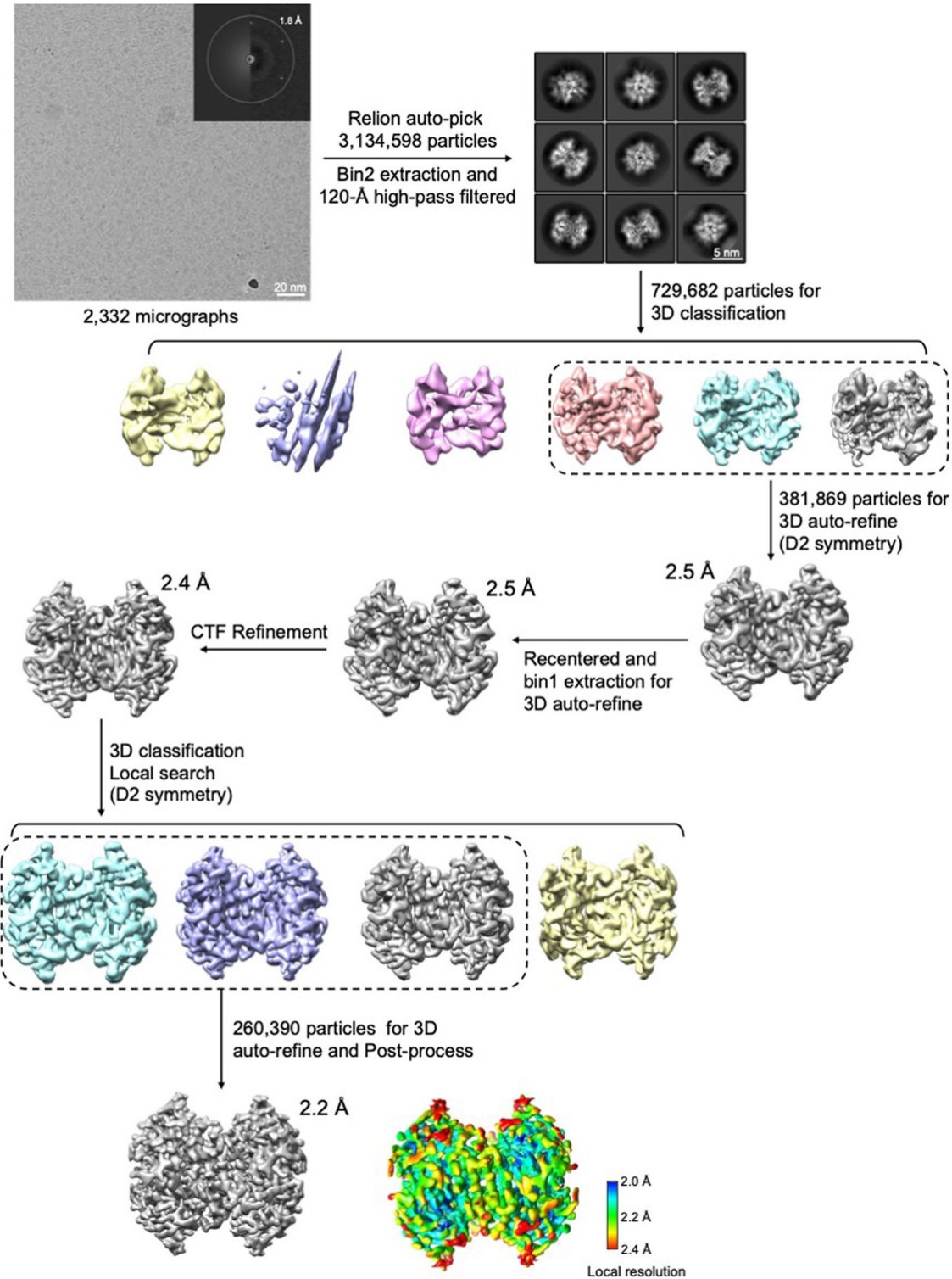

**Extended Data Fig. 10 | The Cryo-EM data-processing workflow of streptavidin (52 kDa) supported by the UFG grid.**

# Reporting Summary

Nature Research wishes to improve the reproducibility of the work that we publish. This form provides structure for consistency and transparency in reporting. For further information on Nature Research policies, see our Editorial Policies and the Editorial Policy Checklist.

## Statistics

For all statistical analyses, confirm that the following items are present in the figure legend, table legend, main text, or Methods section.

| n/a | Confirmed | |
|---|---|---|
| ☐ | ☒ | The exact sample size (*n*) for each experimental group/condition, given as a discrete number and unit of measurement |
| ☐ | ☒ | A statement on whether measurements were taken from distinct samples or whether the same sample was measured repeatedly |
| ☒ | ☐ | The statistical test(s) used AND whether they are one- or two-sided<br>*Only common tests should be described solely by name; describe more complex techniques in the Methods section.* |
| ☒ | ☐ | A description of all covariates tested |
| ☒ | ☐ | A description of any assumptions or corrections, such as tests of normality and adjustment for multiple comparisons |
| ☐ | ☒ | A full description of the statistical parameters including central tendency (e.g. means) or other basic estimates (e.g. regression coefficient) AND variation (e.g. standard deviation) or associated estimates of uncertainty (e.g. confidence intervals) |
| ☒ | ☐ | For null hypothesis testing, the test statistic (e.g. *F*, *t*, *r*) with confidence intervals, effect sizes, degrees of freedom and *P* value noted<br>*Give P values as exact values whenever suitable.* |
| ☒ | ☐ | For Bayesian analysis, information on the choice of priors and Markov chain Monte Carlo settings |
| ☒ | ☐ | For hierarchical and complex designs, identification of the appropriate level for tests and full reporting of outcomes |
| ☒ | ☐ | Estimates of effect sizes (e.g. Cohen's *d*, Pearson's *r*), indicating how they were calculated |

*Our web collection on statistics for biologists contains articles on many of the points above.*

## Software and code

Policy information about availability of computer code

| | |
|---|---|
| Data collection | We used ABAQUS 2017 software to perform finite element simulations of graphene rippling. We used AutoEMation software to collect single-particle cryo-EM datasets, written by Dr. Jianlin Lei at Tsinghua University. We used SerialEM (version 3.8) software to collect tilt series. |
| Data analysis | We used MotionCor2 (version 1.1.0) to correct the beam-induced motion of cryo-EM micrographs and used Relion (version 3.1.3) to perform 3D reconstruction. The CTF values of these motion-corrected micrographs were determined by CTFFIND4 algorithm (version 4.15). The structural analysis was performed in UCSF Chimera (version 1.13.1). The coordinate was generated in PHENIX (version1.14-3260). For cryo-ET reconstruction, we used Etomo (version 4.11) to align and reconstruct the tomograms. All these softwares are open-source. All these softwares are open-source. |

For manuscripts utilizing custom algorithms or software that are central to the research but not yet described in published literature, software must be made available to editors and reviewers. We strongly encourage code deposition in a community repository (e.g. GitHub). See the Nature Research guidelines for submitting code & software for further information.

## Data

Policy information about availability of data

All manuscripts must include a data availability statement. This statement should provide the following information, where applicable:
- Accession codes, unique identifiers, or web links for publicly available datasets
- A list of figures that have associated raw data
- A description of any restrictions on data availability

Data supporting the findings in this manuscript are available from the corresponding authors upon reasonable requests. The coordinates and density maps of streptavidin, hemoglobin and alpha-fetoprotein have been deposited in the RCSB Protein Data Bank (PDB) under the accession number 8GVK, 7XGY and 7YIM, and the EMDB under accession number EMD-32099, EMD-33189 and EMD-33861, respectively. The raw cryo-EM dataset of streptavidin, containing the unaligned

movies and particles, has been deposited into EMPIAR under accession number EMPIAR-11217. Source data of Fig. 2c, Fig. 3b and c, Fig. 4b, d, f, h, i, j and k, Extended Data Fig. 1c and f, Extended Data Fig. 2e, Extended Data Fig. 3b and e, Extended Data Fig. 4g and h, Extended Data Fig. 5c-f, Extended Data Fig. 8, and Extended Data Fig. 9a-c are provided as Source Data files.

# Field-specific reporting

Please select the one below that is the best fit for your research. If you are not sure, read the appropriate sections before making your selection.

☒ Life sciences ☐ Behavioural & social sciences ☐ Ecological, evolutionary & environmental sciences

For a reference copy of the document with all sections, see nature.com/documents/nr-reporting-summary-flat.pdf

# Life sciences study design

All studies must disclose on these points even when the disclosure is negative.

| | |
|---|---|
| Sample size | For cryo-EM analysis, the particle numbers used for the final reconstructions of hemoglobin, alpha-fetoprotein and streptavidin were 105,000, 354,264 and 260,390, respectively, and the reported resolutions were 3.5 Å, 2.6 Å and 2.2 Å, respectively, estimated by the Fourier Shell Correction (FSC)=0.143 cutoff criteria. In cryo-EM field, particle number used for final structural reconstruction varies greatly, which is well acceptable for such sample size used here.<br>To plot and compare the distribution of defocus ranges, we used 313 UFG-supported micrographs and 324 RG-supported micrographs. For cryo-EM reconstruction of 20S proteasome, we normally collect hundreds of micrographs, which are enough for us to get a reconstruction at near-atomic resolution and analyze the defocus ranges.<br>To measure the particle motion, we carried out three particle-motion measurements of both UFG and RG, every of which was based on thousands of particles. Such example size and times of repeated experiments have been widely used to characterize beam-induced motion. |
| Data exclusions | For cryo-EM reconstruction, particles grouped in bad classes with poorly defined features were excluded, because these particles were normally denatured or dissociated samples, which were harmful for high-resolution 3D reconstruction. |
| Replication | Particle-motion measurements were repeated 3 times (as indicated in Methods), and all attempts at replication were successful with similar results. |
| Randomization | Samples were allocated random, including the particle-motion and structural determination in Relion (version 3.1.3). |
| Blinding | For cryo-EM reconstruction in Relion (version 3.1.3), particles were randomly divided into two parts, and used for 3D structure determination. The consistence of structures generated by these two sub-datasets was used for the blinding test. The investigators were blinded to group allocation during data collection and/or analysis. |

# Reporting for specific materials, systems and methods

We require information from authors about some types of materials, experimental systems and methods used in many studies. Here, indicate whether each material, system or method listed is relevant to your study. If you are not sure if a list item applies to your research, read the appropriate section before selecting a response.

## Materials & experimental systems

| n/a | Involved in the study |
|---|---|
| ☒ | Antibodies |
| ☒ | Eukaryotic cell lines |
| ☒ | Palaeontology and archaeology |
| ☒ | Animals and other organisms |
| ☒ | Human research participants |
| ☒ | Clinical data |
| ☒ | Dual use research of concern |

## Methods

| n/a | Involved in the study |
|---|---|
| ☒ | ChIP-seq |
| ☒ | Flow cytometry |
| ☒ | MRI-based neuroimaging |

