## [Peer Review File · Nature Methods]

Peer Review Information

Manuscript Title: Uniform thin ice on ultraflat graphene for high-resolution cryo-EM

Corresponding author name(s): Xiaoding Wei, Hong-Wei Wang, Nan Liu, Hailin Peng

Editorial Notes:

Reviewer Comments & Decisions:

Decision Letter, initial version:
--

Dear Hongwei,

Thank you for your letter detailing how you would respond to the reviewer concerns regarding your Article, "Uniform thin ice on ultraflat graphene for high-resolution cryo-EM". We have decided to invite you to revise your manuscript as you have outlined, before we reach a final decision on publication. I am glad to hear that you are willing to include additional demonstrations. I would like to emphasize though that a challenging case will make your paper stronger; I am happy to discuss this further if you like.

- * include a point-by-point response to the reviewers and to any editorial suggestions
- * please underline/highlight any additions to the text or areas with other significant changes to facilitate review of the revised manuscript
- * address the points listed described below to conform to our open science requirements

* ensure it complies with our general format requirements as set out in our guide to authors at www.nature.com/naturemethods

* resubmit all the necessary files electronically by using the link below to access your home page

[Redacted] This URL links to your confidential home page and associated information about manuscripts you may have submitted, or that you are reviewing for us. If you wish to forward this email to co-authors, please delete the link to your homepage.

If you cannot send it within this time, please let us know. In this event, we will still be happy to reconsider your paper at a later date so long as nothing similar has been accepted for publication at Nature Methods or published elsewhere.

OPEN SCIENCE REQUIREMENTS

REPORTING SUMMARY AND EDITORIAL POLICY CHECKLISTS

IMAGE INTEGRITY

DATA AVAILABILITY

Please include a “Data availability” subsection in the Online Methods. This section should inform readers about the availability of the data used to support the conclusions of your study, including accession codes to public repositories, references to source data that may be published alongside the paper, unique identifiers such as URLs to data repository entries, or data set DOIs, and any other statement about data availability. At a minimum, you should include the following statement: “The data that support the findings of this study are available from the corresponding author upon request”, describing which data is available upon request and mentioning any restrictions on availability. If DOIs are provided, please include these in the Reference list (authors, title, publisher (repository name), identifier, year). For more guidance on how to write this section please see: <http://www.nature.com/authors/policies/data/data-availability-statements-data-citations.pdf>

CODE AVAILABILITY

Please include a “Code Availability” subsection in the Online Methods which details how your custom code is made available. Only in rare cases (where code is not central to the main conclusions of the paper) is the statement “available upon request” allowed (and reasons should be specified).

MATERIALS AVAILABILITY

SUPPLEMENTARY PROTOCOL

To help facilitate reproducibility and uptake of your method, we ask you to prepare a step-by-step Supplementary Protocol for the method described in this paper. We [encourage authors to share their step-by-step experimental protocols](https://www.nature.com/nature-research/editorial-policies/reporting-standards#protocols) on a protocol sharing platform of their choice and report the protocol DOI in the reference list. Nature Research's Protocol Exchange is a free-to-use and open resource for protocols; protocols deposited in Protocol Exchange are citable and can be linked from the published article. More details can found at www.nature.com/protocolexchange/about.

ORCID

Sincerely,
Arunima

Arunima Singh, Ph.D.
Senior Editor
Nature Methods

Reviewers' Comments:

Reviewer #1:

Remarks to the Author:

CryoEM is a blooming field in molecular structural biology. A remaining challenge for some specimens is to generate a uniformly thin and flat ice to embed the particles of interest. The hypothesis of this study is that the suitably thin and flat ice without wrinkles is critical to prepare cryo-specimen of single particles making them more amenable for atomic resolution structure determination. This study introduced a method to prepare a thin graphene substrate grown on a Cu (111) /sapphire wafer with subnanometer rather than tens of nanometer wrinkles. They also carried out extensive mechanical characterizations of the thin and flat graphene. The most convincing evidence to support their hypothesis is the application of their method to embed streptavidin, a 52 kDa protein and to solve its structure at 2.2 Å. Their result breaks the current resolution record for such small protein at closer to atomic resolution. The weakness of the manuscript is to have only one example of small protein and it is not clear on the reproducibility to other macromolecules. Although its general effectiveness is yet to be demonstrated, their hypothesis and methods of investigations are sound and novel. This study would appeal to cryoEM researchers and is worthy for publication. The followings are my comments.

1. It will be helpful to state the ease of preparing such support film for biologists who do not have sophisticated thin film preparative laboratory. It is perfectly ok to acknowledge if the method is difficult to carry out. The result obtained by these authors is novel and encouraging to inspire further development making it more accessible if not so at the present time.
2. Can the authors comment on the reproducibility of the methodology? How many times did they repeat the experiments to get similarly thin ice sample as judged by tilting expt?
3. Why did the authors not apply to another small protein to demonstrate the general applicability of the method?

4. It would be necessary to include a supplementary figure to show the workflow of the image processing (number of initial and final number of particles, the extensiveness of particle classification and quantification of the resolvability of the map. How do the authors know

Reviewer #2:

Remarks to the Author:

Key results

Sample preparation remains a key rate-limiting step in cryo-EM single particle analysis (SPA). Macromolecules need to be suspended in a thin layer of ice to minimize electron scattering by the ice. Uneven ice or large areas of thick ice across a grid – typically produced by currently available plunge freezing methods – can reduce the efficiency of data collection. The use of graphene-derived support films for SPA has been in use since at least 2010, beginning with graphene oxide and more recently with monolayer graphene as alternatives to amorphous carbon films. Graphene support films can be used to capture macromolecules that prefer binding to carbon and reduce deleterious interactions with the air-water interface, while remaining relatively electron transparent at 300 kV.

In this work, Zheng et al. show that the existing method for preparing graphene grids results in rough graphene (RG) with 10~20 nm variations in surface height, and they develop a method for producing ultraflat graphene (UFG) cryo-EM grids with ~2 nm variations in surface height. The authors then use a combination of atomic force microscopy (AFM), scanning EM (SEM), simulations, tomography, and single particle analysis to characterize the performance of their UFG vs RG grids.

Validity

The authors have performed a thorough and interesting characterization of graphene surface structure on cryo-EM grids. By using AFM and modeling data, their characterization of the structure of UFG vs RG grids is convincing. We also enjoyed the combined use of tomography and SPA data to characterize the positions of particles on the surface of the graphene, and therefore also the surface structure of the underlying graphene. The data on the physical properties of UFG and the data from the SPA reconstructions nicely agree with the existing literature.

Significance

Unfortunately, we are not convinced this study represents a significant advance of global interest to the field. The problem of RG's variation in surface height, which this study improves on by using UFG, is not a limiting factor for SPA. While a Guinier plot B-factor of 55 Å² with UFG is good, RG also performs well at B=79 Å². SPA on streptavidin with RG by Han et al. in 2019 (PNAS) produced a 2.6-Å reconstruction with far fewer particles than this paper. Furthermore, local variations in particle height can be corrected

for at the post-processing stage. Indeed, this must be done whether one chooses to use RG or UFG, as the nm-scale variations in UFG height still necessitate such corrections.

Data, methodology, and analysis

- Would the authors comment on the feasibility of producing or obtaining UFG deposited on Cu(111)/sapphire? A novel method needs also to be accessible to the community. What are the costs, time requires etc? Does this differ significantly from from RG films?
- It is not clear to me what pre-tension is. Where does it come from with the UFG? A couple of sentences with a brief background and one or two references could help clarify this for a non-materials person such as myself.
- In the methods section, the authors should describe in detail the glow discharging conditions used to render the graphene grids hydrophilic.
- Since several datasets were collected, a table describing the SPA data collection and processing parameters for the proteasome and streptavidin on UFG and RG would be useful for comparing conditions.
- Since the grid/graphene/Cu/sapphire composite was submerged in APS, was a special grid type required? Regular Quantifoil grids with copper bars will presumably also have their bars etched, otherwise? Please specify the grid types that were used with this method.
- The surface of the holey grid foil is also not perfectly flat. Would the authors be able to share their experience or thoughts on how variations in the grid foil height affect (or do not affect) the flatness of the graphene layer upon stitching?

Clarity and context

- The 'Etching' section of 'Face-to-face transfer of graphene onto TEM grids' methods section is confusing: the graphene seems to be combined with the grids before etching in 'Face-to-face transfer', yet in lines 335-336 there is a sentence saying that the graphene film was transferred onto the grids after etching. Please clarify.
- A figure illustrating the method for face-to-face transfer of graphene onto TEM grids could be useful.

Reviewer #3:

Remarks to the Author:

This ms describes a very exciting advance in cryo-EM specimen technology. Graphene grown on an ultra-flat surface turns out to be pre-tensioned and, due to its flatness, produces more uniform ice thickness, less beam-induced motion and much better results.

The work is of high quality and is an exhaustive characterization of the new cryo-EM substrate.

Major comments.

1. Why is there nonzero tension in the UFG film? Does this have to do with lattice mismatch and epitaxial graphene growth, or is some other origin of it?
2. It appears that you are using conventional holey-carbon grids, rather than all-gold grids with a holey gold film. Please point this out. Which metal were the grids made of?
3. The reduced motion (Fig. 3J) is a surprising result, in view of the otherwise ordinary grids being used. The mismatch of thermal expansion of the metal grid and the honey carbon film, for example, seem not to be so important after all?
4. Meanwhile in Fig. 3J the initial beam-induced motion (during the first 2-3 e/A²) seems to be the same. Why do you think this is?
5. There are very many errors in English usage, only some of which are listed below.

Minor comments

1. What is the meaning of LG and SG in column 1 of Extended Table 1?
2. Do you really mean "atomic" step in line 96?
3. What is meant by '2 1/nm' in line 655? Maybe 2 nm⁻¹ would be better?
4. Some repeating problems in English usage
'Method' e.g. in line 100 should be 'Methods'
'Extend' e.g. in line 89 should be 'Extended'
'waved' e.g. in line 140 should be 'wavy' or maybe a fancier word could be used. Undulating??
'compacted by' in line 200 should be 'composed of'
'slicers' in line 538 should be 'slices'

Author Rebuttal to Initial comments
--

Response to the 1st Reviewer

Cryo-EM is a blooming field in molecular structural biology. A remaining challenge for some specimens is to generate a uniformly thin and flat ice to embed the particles of interest. The hypothesis of this study is that the suitably thin and flat ice without wrinkles is critical to prepare cryo-specimen of single particles making them more amenable for atomic resolution structure determination. This study introduced a method to prepare a thin graphene substrate grown on a Cu

(111)/sapphire wafer with subnanometer rather than tens of nanometer wrinkles. They also carried out extensive mechanical characterizations of the thin and flat graphene. The most convincing evidence to support their hypothesis is the application of their method to embed streptavidin, a 52 kDa protein and to solve its structure at 2.2 Å. Their result breaks the current resolution record for such small protein at closer to atomic resolution. The weakness of the manuscript is to have only one example of small protein and it is not clear on the reproducibility to other macromolecules. Although its general effectiveness is yet to be demonstrated, their hypothesis and methods of investigations are sound and novel. This study would appeal to cryo-EM researchers and is worthy for publication. The followings are my comments.

Response:

We deeply appreciate the positive and insightful comments from the reviewer on the innovation of our work. The reviewer's constructive suggestions help bring significant improvements to our manuscript.

To demonstrate the general effectiveness of the method, we reconstructed the cryo-EM structure of other small macromolecules using ultraflat graphene (UFG) membranes, including the human alpha-fetoprotein (67 kDa) with no symmetry at 2.6 Å resolution, whose other homologous structures were all determined by X-ray crystallography before (Handing, K. B. *et al. Molecular Immunology*. **2016**, *71*, 143-151; Bujacz, A. *et al. Acta Crystallogr D Struct Biol*. **2017**, *73*, 896-909), likely due to its small molecular weight and no symmetry, and hemoglobin (64 kDa) at 3.5 Å, wherein the side chains and bound ligands could be unambiguously recognized. We will fully address the reviewer's comments point by point in the following.

1. It will be helpful to state the ease of preparing such support film for biologists who do not have sophisticated thin film preparative laboratory. It is perfectly ok to acknowledge if the method is difficult to carry out. The result obtained by these authors is novel and encouraging to inspire further development making it more accessible if not so at the present time.

Response:

We appreciate the insightful and kind comments by the reviewer. We agree that the accessibility of graphene film is a practical barrier to the wide application of graphene in the EM community, although the growth of high-quality large-area graphene film is relatively mature in the materials science community (Figure 1a and 1d, Lin L. *et al. Chem. Rev.* **2018**, *118*, 9281). Nowadays, both wafer-scale UFG wafer on Cu(111)/sapphire and large-area graphene film on copper foil have been commercialized, and can be obtained from the market, for example, Beijing Graphene Institute (BGI) Company (<http://www.bgi-graphene.com/type/100>; Email: sales@bgi-graphene.com; Tel. 086-10-83432613) and Graphenea company (<https://www.graphenea.com/>).

To help biologists who do not have thin film preparative laboratory to better understand our strategy and prepare the home-made ultraflat support film, we have provided detailed information,

including the make and model of home-made chemical vapor deposition (CVD) system in the Extended Data Fig. 20, for its possible set-up in biological EM labs. The copper/sapphire wafers can be fabricated using a common magnetron sputtering technique with the help of a nanofabrication technology center in universities or companies in the semiconductor industry. For the growth of UFG on copper/sapphire substrate, the CVD system is the key to reliable graphene film growth with provided growth parameters. All the details have been included in the step-by-step protocol within Methods Section.

Extended Data Fig. 20 | Make and model of the CVD system. The home-made CVD system typically consists of gas supply system (high-purity compressed gas, JINGHUI GAS), control system (mass flow controller, S48 32/HMT, HORIBAR METRON), CVD reactor (tube furnace, Thermo SCIENTIFIC; Quartz tube, 120×1500 mm) and pump system (pump, GLD-N051, ULVAC Inc).

2. Can the authors comment on the reproducibility of the methodology? How many times did they repeat the experiments to get similarly thin ice sample as judged by tilting expt?

Response:

The UFG CVD growth and grid preparation are quite reproducible. We have included the reproducibility of the methodology of UFG grid preparations in the supporting information. The tilting experiments have been repeated more than 10 times, and we performed the tomogram reconstruction to calculate the ice thickness (as Figure 3 f-g did), all demonstrating ~20-nm ice thickness (Figure R1).

Figure R1. Ice thickness calculation based on reconstructed tomograms.

Furthermore, we also collected more than 30 pairs of micrographs across the UFG grids under with and without energy filter slit (20eV slit width) conditions to characterize the ice thickness, based on the formula: $d = \Lambda \ln \frac{I_o}{I_{zlp}}$ (Rice, W. J. *et al. J. Struct. Biol.* **2018**, *204*, 38-44), where I_o is the integrated total intensity, I_{zlp} is the integrated zero-loss peak intensity with energy filter used, and Λ is the apparent mean free path for inelastic scattering (~ 400 nm) (Yonekura, K. *et al. J. Struct. Biol.* **2006**, *156*, 524–536). The ice thickness distribution was plotted below (Figure R2), consistent with the cryo-ET characterization results (~ 20 -nm ice thickness).

Figure R2. Ice thickness calculation using an energy filter with a slit width of 20 eV.

3. Why did the authors not apply to another small protein to demonstrate the general applicability of the method?

Response:

We are very thankful for the constructive comments. In addition to the streptavidin (52 kDa), we have applied the UFG grids for obtaining the structure of hemoglobin (64 kDa) at 3.5 Å resolution with the standard cryo-EM data processing workflow, wherein the side chains and bound ligands could be unambiguously recognized (Figure R3). Moreover, we also used UFG grids to solve the structure of human alpha-fetoprotein (67 kDa) at 2.6 Å resolution (Figure R4), a serum albumin protein with no symmetry, whose other homologous structures were all determined by X-ray crystallography before, likely due to its small molecular weight and no symmetry. We collected one-day cryo-EM datasets (2,290 micrographs) and followed the standard data-processing workflow to get a 2.6-Å resolution reconstruction. All the information has been included in the revised Figure 4, Extended Data Fig. 15, and Extended Data Fig. 16 (Page 20 and Page 34 of manuscript). In addition, the UFG grids have been used by other collaborative laboratories for cryo-EM structural determination with protein amassing <100 kDa.

Figure R3. Cryo-EM reconstruction of hemoglobin (64 kDa) supported by UFG membrane. **a**, A representative cryo-EM micrograph of hemoglobin on UFG membrane. **b**, The Gold-Standard

Fourier Shell Correlation (FSC) curve. FSC=0.143 was indicated by the dotted line, used for estimating the resolution (~ 3.5 Å). **c**, The density map of hemoglobin, with individual monomers colored by different colors. **d**, The selected densities with corresponding coordinates docked.

Figure R4. Cryo-EM reconstruction of human alpha-fetoprotein (67 kDa) supported by UFG membrane. **a**, A representative cryo-EM micrograph of alpha-fetoprotein on UFG membrane. **b**, The two-dimensional classification results of alpha-fetoprotein. **c**, The density map of alpha-fetoprotein. **d**, The Gold-Standard Fourier Shell Correlation (FSC) curve. FSC=0.143 was indicated by the blue line, used for estimating the resolution (2.6 Å). **e**, The selected densities with corresponding coordinates docked.

4. It would be necessary to include a supplementary figure to show the workflow of the image processing (number of initial and final number of particles, the extensiveness of particle classification and quantification of the resolvability of the map. How do the authors know.

Response:

We are very thankful for the reviewer's comments. The cryo-EM data processing workflow has been included in the Extended Data Fig. 18 (Page 36 of the manuscript), as is shown below:

Extended Data Fig. 18 | The Cryo-EM data-processing workflow of streptavidin (52 kDa) supported by the UFG grid.

To better compare the cryo-EM data processing parameters, we also added a summary table (Extended Table 2) describing the data collection and processing for the proteasome, streptavidin, hemoglobin, and alpha-fetoprotein, as below:

Extended Table 2: Summary of cryo-EM data collection and processing

	20S proteasome	20S proteasome	Hemoglobin	Alpha-fetoprotein	Streptavidin
Molecular weight (kDa)	700	700	64	67	52
Supporting film	Rough Graphene (RG)	Ultraflat Graphene (UFG)	Ultraflat Graphene (UFG)	Ultraflat Graphene (UFG)	Ultraflat Graphene (UFG)
Magnification	29,000	29,000	165,000	165,000	165,000
Voltage (kV)	300	300	300	300	300
Pixel size (Å)	0.97	0.97	0.5191	0.5191	0.5191
Electron exposure (e ⁻ /Å ²)	50	50	50	50	50
Defocus range (µm)	0.6-2.5	0.5-2.3	0.6-2.4	0.6-2.5	0.7-2.2
Symmetry imposed	D7	D7	C2	C1	D2
Micrographs (no.)	324	316	5,604	2,290	2,332
Initial particle images (no.)	150,774	143,954	5,761,072	1,660,932	3,134,598
Final particle. images (no.)	12,978	7,152	105,000	354,264	260,390
Resolution (Å)	3.0	2.8	3.5	2.6	2.2

Response to the 2nd Reviewer

Key results

Sample preparation remains a key rate-limiting step in cryo-EM single particle analysis (SPA). Macromolecules need to be suspended in a thin layer of ice to minimize electron scattering by the ice. Uneven ice or large areas of thick ice across a grid—typically produced by currently available plunge freezing methods—can reduce the efficiency of data collection. The use of graphene-derived support films for SPA has been in use since at least 2010, beginning with graphene oxide and more recently with monolayer graphene as alternatives to amorphous carbon films. Graphene support films can be used to capture macromolecules that prefer binding to carbon and reduce deleterious interactions with the air-water interface, while remaining relatively electron transparent at 300 kV.

In this work, Zheng et al. show that the existing method for preparing graphene grids results in rough graphene (RG) with 10~20 nm variations in surface height, and they develop a method for producing ultraflat graphene (UFG) cryo-EM grids with ~2 nm variations in surface height. The authors then use a combination of atomic force microscopy (AFM), scanning EM (SEM), simulations, tomography, and single particle analysis to characterize the performance of their UFG vs RG grids.

Validity

The authors have performed a thorough and interesting characterization of graphene surface structure on cryo-EM grids. By using AFM and modeling data, their characterization of the structure of UFG vs RG grids is convincing. We also enjoyed the combined use of tomography and SPA data to characterize the positions of particles on the surface of the graphene, and therefore also the surface structure of the underlying graphene. The data on the physical properties of UFG and the data from the SPA reconstructions nicely agree with the existing literature.

Response:

We sincerely appreciate the positive comments from the reviewer on the results and validity of our work. The reviewer's constructive suggestions help bring significant improvements to our manuscript.

Significance

Unfortunately, we are not convinced this study represents a significant advance of global interest to the field. The problem of RG's variation in surface height, which this study improves on by using UFG, is not a limiting factor for SPA. While a Guinier plot B-factor of 55 Å² with UFG is good, RG also performs well at B=79 Å². SPA on streptavidin with RG by Han et al. in 2019 (PNAS) produced a 2.6-Å reconstruction with far fewer particles than this paper. Furthermore,

local variations in particle height can be corrected for at the post-processing stage. Indeed, this must be done whether one chooses to use RG or UFG, as the nm-scale variations in UFG height still necessitate such corrections.

Response:

We thank the referee for raising this concern. We think the UFG supports have the following outstanding advances for global interest to the field:

- 1) Ultraflat support and uniform ice are very useful or even indispensable for high-quality tilt series data collection (Scheme 1 and Figure 3 a-h).
- 2) The contrast (signal-to-noise ratio) could be severely impaired by the 10~20 nm variation ice thickness for small molecules, which introduces uncertainties during the post-processing correction of local variation. The UFG enables uniform thin ice and thus ensures high-quality data with more homogenous local CTF and a better signal-to-noise ratio at the data-collection stage.
- 3) In many high-resolution cryo-EM studies, large amounts of data (as much as 95% or more of the grids, micrographs, or particles) are discarded during data collection or processing (C. J. Russo *et al. Journal of Structural Biology* **2016**, *193*, 33). Therein, the ice thickness variation is often considered one of the key factors. For the structure determination of streptavidin on RG, the final particle number used in the previous work that the referee mentioned (2.6-Å reconstruction) is 11,402 with an initial particle number of 1,130,061. The data utilization efficiency on RG is 1.0%. In comparison, the final particle number used on UFG is 260,390 with a final reconstruction resolution of 2.2 Å, selected from the initial particles with a number of 3,134,598. The data utilization efficiency is 8.3%. Moreover, we are able to obtain a 2.7-Å reconstruction of streptavidin using only 3,358 particles on UFG. Thus, the UFG significantly increased the data utilization efficiency and enabled much higher resolution.
- 4) In term of graphene grids preparation (Figure 1f and Extended Data Fig. 4), the flat surface of UFG enabled better interfacial contact with EM grids during the face-to-face transfer procedure, thus making the graphene transfer process more controllable and giving rise to a higher fabrication rate of graphene grids, compared with RG.
- 5) The preparation of UFG has become the leading-edge field in materials science (Wang, M. *et al. Nature* **2021**, *596*, 519. Yuan, G. W. *et al. Nature* **2020**, *577*, 204). The design of suspended UFG will be generalized to other two-dimensional materials and will be further extended to the applications of nanoelectromechanical system (NEMS) devices, photoelectric devices, and separation membranes (X. Fan *et al., Nat. Electron.* **2019**, *2*, 394. Y. Kim *et al., Nat. Nanotech.* **2015**, *10*, 676. P. Kidambi, *et al. Science* **2021**, *374*, 708).

Data, methodology, and analysis

- Would the authors comment on the feasibility of producing or obtaining UFG deposited on Cu(111)/sapphire? A novel method needs also to be accessible to the community. What are the costs, time requires etc? Does this differ significantly from RG films?

Response:

We thank the referee for raising this concern. The growth of high-quality large-area graphene film is relatively mature in materials science (Figure 1a and 1d; Lin L. *et al. Chem. Rev.* **2018**, *118*, 9281). Nowadays, both wafer-scale UFG wafer on Cu(111)/sapphire and large-area graphene film on copper foil have been commercialized, and can be obtained from the market, for example, Beijing Graphene Institute (BGI) Company (<http://www.bgi-graphene.com/type/100>; Email: sales@bgi-graphene.com; Tel. 086-10-83432613) and Graphenea company (<https://www.graphenea.com/>).

The copper/sapphire wafers can be fabricated using a common magnetron sputtering technique with the help of nanofabrication technology center in the universities or companies in the semiconductor industry, whose cost, in our practice, is ~ \$1700 for 25 pieces of 4-inch copper/sapphire wafers (i.e., the average cost of each 4-inch copper/sapphire wafer is ~ \$68). During the face-to-face graphene transfer procedure, a 4-inch UFG wafer can produce at least 600 pieces of UFG grids, adding ~ \$0.11 more cost for each grid.

For the growth of UFG or rough graphene film, the typical chemical vapor deposition (CVD) system is the key to reliable graphene growth with provided growth parameters. Yet, we think the set-up of CVD system might be a primary barrier to the accessibility of graphene film in the cryo-EM community. Thus, we have provided more information on the make and model of a home-made CVD system in the Extended Data Fig. 20 for its possible set-up in biological EM labs. All the experimental and time-cost details have been included in the step-by-step protocol within Methods Section. Note that the cost and time required for UFG wafer growth were similar to those of typically rough graphene films on copper foils.

Extended Data Fig. 20 | Make and model of the CVD system. The home-made CVD system typically consists of gas supply system (high-purity compressed gas, JINGHUI GAS), control system (mass flow controller, S48 32/HMT, HORIBAR METRON), CVD reactor (tube furnace, Thermo SCIENTIFIC; Quartz tube, 120×1500 mm) and pump system (pump, GLD-N051, ULVAC Inc).

- It is not clear to me what pre-tension is. Where does it come from with the UFG? A couple of sentences with a brief background and one or two references could help clarify this for a non-materials person such as myself.

Response:

We are very thankful for the constructive comments. The pre-tension means the tensile stress in the basal plane of the graphene membrane. It originates from the interaction between the sidewall of the holes in the carbon film and monolayer graphene. Since the bending rigidity of graphene monolayers is extremely small, the sidewall of the holes in the carbon film attracts the graphene membrane so that the edge of the membrane will be sucked to the sidewall. This interaction then induces the pre-tension in the graphene membrane. In fact, the graphene membrane being attracted to the sidewalls has also been evidenced by the insets of Extended Data Fig. 7g-h, which shows that the periphery of the suspended UFG is pulled down by 1~2 nm and attached to the sidewall.

This pre-tension in the flat graphene suspended on a holey rigid substrate has been reported in previous literature. Lee, C. and Wei, X. D. et al. firstly measured the mechanical properties of freestanding graphene monolayers and reported a pre-tension was in the range of 0.1 to 0.7 N/m (Lee, C., Wei, X. D. *et al. Science* **2008**, 321, 385). Similar pre-tension in monolayer graphene

drum samples has also been reported by other groups (Bunch, J. S. *et al. Nano Lett.* **2008**, 8, 2458), and reproduced by simulations (Budrikis, Z. *et al. Nano Lett.* **2016**, 16, 387).

In contrast, the rough graphene film is overly relaxed (especially in the direction perpendicular to the wrinkled lines), and the attraction by the sidewalls could hardly introduce a notable tensile stress in the membrane. This is why the measured pre-tensions in rough graphene are around zero and notably lower than those in the UFG (Fig. 2c).

To clarify the origin of the pre-tension, we have accordingly revised page 4 of the manuscript as follows: "...The pre-tension in suspended UFG mainly lies in the fact that the periphery of the graphene membrane is attracted by the sidewalls of the holes in carbon film (insets of Extended Data Fig. 7g-h). The pre-tension induced by the interaction between graphene membrane and the sidewalls of holey substrates has been reported in previous experiments^{27,28} and molecular dynamics simulations²⁹..."

- In the methods section, the authors should describe in detail the glow discharging conditions used to render the graphene grids hydrophilic.

Response:

We are very thankful for the reviewer's suggestions. We have accordingly revised page 10 of the manuscript as follows:

"Glow discharging of graphene grids

To render the hydrophilic graphene grids, graphene grids were glow-discharged for ~12 s using the "low" setting of a plasma cleaner (Harrick, PDC-32G) after the chamber was evacuated for 2 min."

- Since several datasets were collected, a table describing the SPA data collection and processing parameters for the proteasome and streptavidin on UFG and RG would be useful for comparing conditions.

Response:

We are very thankful for the reviewer's suggestions. We have made a summary table of SPA data collection and processing on page 40 of the revised manuscript (Extended Table 2):

Extended Table 2: Summary of cryo-EM data collection and processing

	20S proteasome	20S proteasome	Hemoglobin	Alpha- fetoprotein	Streptavidin
Molecular	700	700	64	67	52

weight (kDa)					
Supporting film	Rough Graphene (RG)	Ultraflat Graphene (UFG)	Ultraflat Graphene (UFG)	Ultraflat Graphene (UFG)	Ultraflat Graphene (UFG)
Magnification	29,000	29,000	165,000	165,000	165,000
Voltage (kV)	300	300	300	300	300
Pixel size (Å)	0.97	0.97	0.5191	0.5191	0.5191
Electron exposure (e-/Å ²)	50	50	50	50	50
Defocus range (µm)	0.6-2.5	0.5-2.3	0.6-2.4	0.6-2.5	0.7-2.2
Symmetry imposed	D7	D7	C2	C1	D2
Micrographs (no.)	324	316	5,604	2,290	2,332
Initial particle images (no.)	150,774	143,954	5,761,072	1,660,932	3,134,598
Final particle images (no.)	12,978	7,152	105,000	354,264	260,390
Resolution (Å)	3.0	2.8	3.5	2.6	2.2

- Since the grid/graphene/Cu/sapphire composite was submerged in APS, was a special grid type required? Regular Quantifoil grids with copper bars will presumably also have their bars etched, otherwise? Please specify the grid types that were used with this method.

Response:

We are very thankful for the reviewer's suggestions. The commercial Au holey carbon grids (Quantifoil, Au-300 or 200 mesh-R1.2/1.3) were used to prepare graphene grids in our work. We agree that the grids with copper bars should be avoided because the copper bars will be etched by the (NH₄)₂S₂O₈ aqueous solution.

We have accordingly revised page 9 of the manuscript as follows: "... The commercial Au holey carbon grids (Quantifoil, Au-300 or 200 mesh-R1.2/1.3) were used to prepare graphene grids. Note that the grids with copper bars should be avoided because the copper bars can be etched by the (NH₄)₂S₂O₈ aqueous solution..."

- The surface of the holey grid foil is also not perfectly flat. Would the authors be able to share their experience or thoughts on how variations in the grid foil height affect (or do not affect) the flatness of the graphene layer upon stitching?

Response:

We are very thankful for the insightful comments. We think the height variations of holey grid foil will slightly increase the roughness of suspended UFG. The average roughness (R_a) of UFG film on Cu(111)/sapphire was ~ 0.28 nm (Figure 1e). When the UFG was transferred onto holey carbon film, the corresponding R_a of suspended graphene increased to ~ 0.7 nm (Extended Data Fig. 7g), which was still flat and comparable to the flatness of commercial silicon wafer ($R_a \sim 0.7$ nm).

Clarity and context

- The 'Etching' section of 'Face-to-face transfer of graphene onto TEM grids' methods section is confusing: the graphene seems to be combined with the grids before etching in 'Face-to-face transfer', yet in lines 335-336 there is a sentence saying that the graphene film was transferred onto the grids after etching. Please clarify.

Response:

Thank you very much for your valuable comment. The graphene film was indeed combined with TEM grids before etching in 'face-to-face transfer'. To make it more straightforward to readers, we have accordingly revised page 9 of the manuscript as follows: "...~~After etching, the ultraflat graphene film was transferred onto the TEM grids.~~ After the copper film was etched, the graphene grids were obtained with UFG covered on the holey carbon film..."

- A figure illustrating the method for face-to-face transfer of graphene onto TEM grids could be useful.

Response:

We are very thankful for the reviewer's valuable suggestion. To make the face-to-face transfer process clearer to readers, we have accordingly added a schematic illustration in Extended Data Fig. 21 as suggested (Page 38 of the manuscript).

Extended Data Fig. 21 | Schematic illustration of face-to-face transfer method. a, UFG film on the Cu(111)/sapphire substrate. **b**, Placing TEM grids onto the UFG/Cu(111)/sapphire. **c**, Dropping isopropanol solution to cover the surface of TEM grids on graphene/Cu(111)/sapphire. **d**, Combined TEM grids/graphene/Cu(111) /sapphire composite after isopropanol evaporation. **e**, Submerged composite in the etching solution to etch the Cu(111) film off. **f**, Graphene-coated TEM grids after being rinsed and dried.

Response to the 3rd Reviewer

This ms describes a very exciting advance in cryo-EM specimen technology. Graphene grown on an ultra-flat surface turns out to be pre-tensioned and, due to its flatness, produces more uniform ice thickness, less beam-induced motion and much better results.

The work is of high quality and is an exhaustive characterization of the new cryo-EM substrate.

Response:

We appreciate the positive comments very much from the reviewer on the innovation and quality of our work. The reviewer's constructive suggestions help bring significant improvements to our manuscript.

Major comments.

1. Why is there nonzero tension in the UFG film? Does this have to do with lattice mismatch and epitaxial graphene growth, or is some other origin of it?

Response:

Thanks for the reviewer's valuable comments. The pre-tension means the tensile stress in the basal plane of the graphene membrane. We think the pre-tension originates from the interaction between the sidewall of the holes in the holey carbon film and monolayer graphene. Since the bending rigidity of graphene monolayers is extremely small, the sidewall of the holes in the carbon film attracts the graphene membrane so that the edge of the membrane will be sucked to the sidewall. This interaction then induces the pre-tension in the graphene membrane. In fact, the graphene membrane being attracted to the sidewalls has also been evidenced by the insets of Extended Data Fig. 7g-h, which shows that the periphery of the suspended UFG is pulled down by 1~2 nm and attached to the sidewall.

This pre-tension in the flat graphene suspended on a holey rigid substrate has been reported in previous literature. Lee, C. and Wei, X. D. et al. firstly measured the mechanical properties of freestanding graphene monolayers and reported a pre-tension was in the range of 0.1 to 0.7 N/m (Lee, C., Wei, X. D. *et al. Science* **2008**, *321*, 385). Similar pre-tension in monolayer graphene drum specimens has also been reported by other groups (Bunch, J. S. *et al. Nano Lett.* **2008**, *8*, 2458), and reproduced by simulations (Budrikis, Z. *et al. Nano Lett.* **2016**, *16*, 387).

In contrast, the rough graphene film is overly relaxed (especially in the direction perpendicular to the wrinkled lines), and the attraction by the sidewalls could hardly introduce a notable tensile stress in the membrane. This is why the measured pre-tensions in rough graphene are around zero and notably lower than those in the UFG (Fig. 2c).

To clarify the origin of the pre-tension, we have accordingly revised page 4 of the manuscript as follows: "...The pre-tension in suspended UFG mainly lies in the fact that the periphery of the

graphene membrane is attracted by the sidewalls of the holes in holey film (insets of Extended Data Fig. 7g-h). The pre-tension induced by the interaction between graphene membrane and the sidewalls of holey substrates has been reported in previous experiments^{27,28} and molecular dynamics simulations²⁹..."

2. It appears that you are using conventional holey-carbon grids, rather than all-gold grids with a holey gold film. Please point this out. Which metal were the grids made of?

Response:

Thanks for the reviewer's valuable suggestions. These TEM grids were made of gold bars with holey carbon films.

To make it clear to readers, we have accordingly revised the page 9 of manuscript as follows: "... The commercial Au holey carbon grids (Quantifoil, Au-300 or 200 mesh-R1.2/1.3) were used to prepare graphene grids. Note that the grids with copper bars should be avoided because the copper bars can be etched by the $(\text{NH}_4)_2\text{S}_2\text{O}_8$ aqueous solution..." And we also added a diagram (Extended Data Fig. 21) in the revised manuscript to illustrate the grids used in our work.

3. The reduced motion (Fig. 3J) is a surprising result, in view of the otherwise ordinary grids being used. The mismatch of thermal expansion of the metal grid and the holey carbon film, for example, seem not to be so important after all?

Response:

We appreciate the insightful comments by the reviewer. As has been pointed out in previous studies (K. Naydenova, P. Jia, C. J. Russo, *Science* **2020**, 370, 223; C. J. Russo *et al.* *Science* **2014**, 346, 1377), the bending of the support foil and the ice layer plays a critical role in specimen motion during beam irradiation. Since the carbon-on-gold support foils used for RG and UFG grids are the same, we believe that the reduced particle motion on UFG grids mainly stems from the less bending of the ice layer (Figure R5).

According to the previous analysis (K. Naydenova, P. Jia, C. J. Russo, *Science* **2020**, 370, 223), the compressive stress will build up in the thin ice owing to the inhibited volume change when the liquid water cools rapidly and turns into vitreous ice. Naydenova *et al.* have shown that the largest

volumetric strain in ice is $\left(\frac{\Delta V}{V}\right)_{\max} \sim 0.06$. Assuming ice as an isotropic linear elastic material, the

radial compressive strain is about $\varepsilon_r = -\frac{1}{3}\left(\frac{\Delta V}{V}\right)_{\max} \sim -0.02$. Taking $E_{\text{ice}} \sim 1$ GPa and $\nu_{\text{ice}} \sim 0.3$ as

the Young's modulus and Poisson's ratio of amorphous ice, respectively, there is a radial

compressive stress of $\sigma_r = \frac{E_{ice}}{1-\nu_{ice}} \varepsilon_r \approx -0.029$ GPa in the ice layer (or equivalent to $N_r \sim -0.58$ N/m in 2D stress form since the ice layer thickness is 20 nm). Note that the negative sign means a compressive stress/strain.

According to the physical theory of ice bending (K. Naydenova, P. Jia, C. J. Russo, *Science* **2020**, *370*, 223), the critical compression for a clamped circular membrane to buckle can be approximated by $N_c = -\frac{14.682E_c h_c^3}{12r^2(1-\nu_c^2)}$ in which E_c , h_c and ν_c are the effective Young's modulus, total thickness, and effective Poisson's ratio of the UFG-ice composite layer. Applying the rule of mixtures, we can estimate $E_c \approx 16.4$ GPa and $\nu_c \sim 0.3$. Therefore, the critical compression for the UFG-ice layer to buckle is approximately $N_c \sim -0.13$ N/m.

For UFG, our nanoindentation tests have shown an average pre-tension of approximately 0.2 N/m. Thus, the maximum effective compressive stress in the UFG-ice composite layer is $-0.58 + 0.2 = -0.38$ N/m. Therefore, the calculations above suggest that although the UFG-ice layer might still buckle eventually, the pre-tension in UFG can effectively delay the buckling, and the maximum compression in ice is also greatly reduced (from -0.58 N/m down to -0.38 N/m).

In contrast, the RG has no pre-tension to compensate for the compression in the ice layer, so the RG-ice composite layer has much less resistance to buckling. This could explain why our UFG grids showed significantly reduced beam-induced motion. Figure R5 below illustrates the different buckling-resistance abilities of these two grids.

Figure R5. Schematic illustration showing the beam-induced buckling of graphene-ice layers. a, The pre-tension in UFG (N_{UFG}) compensates the compression in vitreous ice (N_{ice}) and delays the buckling. **b**, Rough graphene possesses no pre-tension so that the buckling occurs much earlier.

To make the reduced motion clearer to the readers, we have accordingly revised page 5 of the manuscript as follows: "... Moreover, the particle motion on UFG induced by the electron beam

was significantly smaller than that on RG (Fig. 3j), ~~indicating that the ice layer on UFG was more resistant to deformation during imaging~~ owing to the ice layer on pretensioned UFG becoming more resistant to deformation during cryo-EM imaging in accordance to the physical theory of ice bending^{32,33} (detailed discussions about the effect of pre-tension on the particle motion reduction can be found in Methods)....”

32 Naydenova, K., Jia, P. P. & Russo, C. J. Cryo-EM with sub-1 angstrom specimen movement. *Science* **370**, 223-226 (2020).

33 Russo, C. J. & Passmore, L. A. Ultrastable gold substrates for electron cryomicroscopy. *Science* **346**, 1377-1380 (2014).

Besides, we have added a section "**Effect of pre-tension in UFG on the particle motion reduction**" in Methods on page 11 to clarify the cause of reduced particle motion.

4. Meanwhile in Fig. 3J the initial beam-induced motion (during the first 2-3 $e^-/\text{\AA}^2$) seems to be the same. Why do you think this is?

Response:

We appreciate the insightful comments by the reviewer. The beam-induced motion generally exhibits two phases: a faster phase during the first 3 $e^-/\text{\AA}^2$ followed by a slower phase (Figure 3J). Previous work showed that much motion in the first phase was due to the support foil (C. J. Russo *et al. Science* **2014**, 346, 1377). Since the same carbon-on-gold grids were used to fabricate both RG and UFG grids, the same initial beam-induced motions (during the first 2-3 $e^-/\text{\AA}^2$) likely stem from the supporting carbon foil.

5. There are very many errors in English usage, only some of which are listed below.

Response:

We appreciate the reviewer's comments, which bring significant improvements in our manuscript. We have carefully revised the language flaws in the manuscript as suggested.

Minor comments

1. What is the meaning of LG and SG in column 1 of Extended Table 1?

Response:

The meanings of LG and SG were large-grain graphene and small-grain graphene, respectively. To make the meanings of LG and SG more straightforward to readers, we have accordingly revised the manuscript as follows:

Extended Table 1 | Summary of Young's modulus and mechanical strength of graphene reported in literature.

Graphene	Young's modulus (GPa)	Mechanical strength (GPa)	Hole size (μm)	Reference
Exfoliated	1010 ± 150	130 ± 10	1-1.5	Science 2008 , 321, 385
Exfoliated	897 ± 41	83.6–104.5	0.5-3	Nat. Phys. 2015 , 11, 26
Exfoliated	1060 ± 80	---	1-1.5	ACS Nano 2016 , 10, 1820
CVD	161 ± 147	35	2	Nano Lett. 2011 , 11, 2259
CVD (LG) Large-grain CVD Graphene	1010 ± 50	103	1-1.5	Science 2013 , 340, 1073
CVD (SG) Small-grain CVD Graphene	980 ± 50	98.5		
CVD	540 ± 33	13.6	2.2	ACS Nano 2013 , 7, 1171
CVD	423	28.7	2.2	ACS Nano 2014 , 8, 10246
Rough Graphene (RG)	721.5 ± 139.4	93.1 ± 24.9	1.2	This work
Ultraflat Graphene (UFG)	933.1 ± 171.0	145.0 ± 13.3		

2. Do you really mean "atomic" step in line 96?

Response:

We are very thankful for the comments. To make the expression more precise, we have accordingly revised the manuscript as follows: "...The height difference of ~~atomic~~ steps on graphene wafer is decreased to ~ 1 nm..."

3. What is meant by '2 1/nm' in line 655? Maybe 2 nm^{-1} would be better?

Response:

We have accordingly revised the manuscript as suggested: "...Scale bars, 2 nm^{-1} ~~2-1/nm~~...."

4. Some repeating problems in English usage

'Method' e.g. in line 100 should be 'Methods'

Response:

We have accordingly revised the manuscript as suggested: "...we sought to transfer the UFG onto EM grids using a face-to-face transfer method (details in ~~Methods~~ **Method**)..."

'Extend' e.g. in line 89 should be 'Extended'

Response:

We have accordingly revised the manuscript as suggested. e.g. "...even at a lateral scale of several micrometers (Fig. 1c, ~~Extend~~ **Extended** Data Fig. 2)..."

'waved' e.g. in line 140 should be 'wavy' or maybe a fancier word could be used. Undulating??

Response:

We deeply appreciate the reviewer's constructive suggestion. We agree that 'wavy' is a fancier word. We have accordingly revised the manuscript as suggested:

"...the vitreous ice on the RG showed a ~~waved~~ **wavy** morphology with inconsistent contrast in the cryo-EM image..."

"...Since the uniform and thin ice on UFG eliminated the ~~waved~~ **wavy** features at the high-angle tilt..."

"...Schematic illustration of wrinkled rough graphene resulting in the ~~waved~~ **wavy** ice with inconsistent ice thickness..."

"...Right: cryo-EM images showing the uniform vitreous ice on the UFG (**f**) and ~~waved~~ **wavy** vitreous ice on the rough graphene (**g**), respectively..."

Besides, we also replaced the word 'waved' with 'wavy' in Scheme 1a, Figure 2g, Figure 3b and Extended Data Fig. 12b, respectively.

'compacted by' in line 200 should be 'composed of'

Response:

We have accordingly revised the manuscript as suggested: "...Streptavidin ~~was compacted by~~ **is composed of** four ~~avidin~~ monomers..."

'slicers' in line 538 should be 'slices'

Response:

We have accordingly revised the manuscript as suggested:

"...Three selected ~~slicers~~-slices through the Z-axis from the tomogram of UFG supported cryo-specimen..."

"...Three selected ~~slicers~~-slices through the Z-axis from the tomogram of RG supported cryo-specimen..."

Decision Letter, first revision:

Dear Hongwei,

Thank you for submitting your revised manuscript "Uniform thin ice on ultraflat graphene for high-resolution cryo-EM" (NMETH-A47297A). It has now been seen by the original referees and their comments are below. The reviewers find that the paper has improved in revision, and therefore we'll be happy in principle to publish it in Nature Methods, pending minor revisions to satisfy the referees' final requests and to comply with our editorial and formatting guidelines.

TRANSPARENT PEER REVIEW

Nature Methods offers a transparent peer review option for new original research manuscripts submitted from 17th February 2021. We encourage increased transparency in peer review by publishing the reviewer comments, author rebuttal letters and editorial decision letters if the authors agree. Such peer review material is made available as a supplementary peer review file. Please state in the cover letter 'I wish to participate in transparent peer review' if you want to opt in, or 'I do not wish to participate in transparent peer review' if you don't. Failure to state your preference will result in delays in accepting your manuscript for publication.

Thank you again for your interest in Nature Methods Please do not hesitate to contact me if you have any questions.

Sincerely,
Arunima

Arunima Singh, Ph.D.
Senior Editor
Nature Methods

ORCID

Reviewer #1 (Remarks to the Author):

The authors have revised the manuscript according to the reviewers comments. The method is well described throughout. I am impressed by the two additional biological examples provided. I would like to suggest that the coordinates for the streptavidin should also be deposited to the PDB. Though the authors are willing to release the raw images, they are encouraged to deposit the raw image data to EMPIAR because they will be beneficial to the community who would like to utilize their method.

Reviewer #2 (Remarks to the Author):

We thank the authors for their very comprehensive and thoughtful responses to all the reviewer comments. Their interdisciplinary work applying advances in material sciences to cryo-EM is the type of work that is critical for addressing the problems in sample preparation that cryo-EM currently faces. The authors have strengthened their work with additional proof of applicability of ultraflat graphene (UFG) to other small (<100 kDa) proteins, and with more comprehensive descriptions and figures from the material sciences perspective. I'm still not 100% sold that ultraflat graphene is really that significant of an advancement over existing graphene technologies, but its applicability to getting small proteins to higher resolutions more efficiently, as the authors have shown, could be important. Overall, our

concerns have been addressed by the authors. We commend the authors on their comprehensive and delightful work.

Reviewer #3 (Remarks to the Author):

The manuscript is much improved, and my questions have been very well answered. I am especially impressed by the reduced beam-induced movement and am grateful for the explanation.

I have two small comments.

1. Line 78, "absorb". I think the word you want is "adsorb" which is quite different. You might need to rephrase the sentence, something like "...target particles can adsorb to the UFG surface..."
2. Line 216-218. The sentence here doesn't make sense to me.
3. One worry about the interaction of proteins with a surface (e.g. your UFG) is the possibility of strong interactions yielding preferred orientations. Clearly you did not have a big problem with this, in view of the beautiful reconstructions, but it would be nice to show in another supplementary figure the map of particle orientations.

-Fred Sigworth

Author Rebuttal, first revision:

Reviewer #1 (Remarks to the Author):

The authors have revised the manuscript according to the reviewers comments. The method is well described throughout. I am impressed by the two additional biological examples provided. I would like to suggest that the coordinates for the streptavidin should also be deposited to the PDB. Though the authors are willing to release the raw images, they are encouraged to deposit the raw image data to EMPIAR because they will be beneficial to the community who would like to utilize their method.

Response: We thank the reviewer for these suggestions, and we have deposited the coordinate of streptavidin to PDB (PDB code: 8GVK) and the raw image data to EMPIAR (EMPIAR-11217).

Reviewer #2 (Remarks to the Author):

We thank the authors for their very comprehensive and thoughtful responses to all the reviewer comments. Their interdisciplinary work applying advances in material sciences to cryo-EM is the type of work that is critical for addressing the problems in sample preparation that cryo-EM currently faces. The authors have strengthened their work with additional proof of applicability of ultraflat graphene (UFG) to other small (<100 kDa) proteins, and with more comprehensive descriptions and figures from the material sciences perspective. I'm still not 100% sold that ultraflat graphene is really that significant of an advancement over existing graphene technologies, but its applicability to getting small proteins to higher resolutions more efficiently, as the authors have shown, could be important. Overall, our concerns have been addressed by the authors. We commend the authors on their comprehensive and delightful work.

Response: We thank the reviewer for his/her recognition of our revision.

Reviewer #3 (Remarks to the Author):

The manuscript is much improved, and my questions have been very well answered. I am especially impressed by the reduced beam-induced movement and am grateful for the explanation.

I have two small comments.

1. Line 78, "absorb". I think the word you want is "adsorb" which is quite different. You might need to rephrase the sentence, something like "...target particles can adsorb to the UFG surface..."

Response: We thank the reviewer for this suggestion, and we have rewritten the sentence as *"Target particles could adsorb onto the UFG surface at the same plane, which keeps them away from the air-water interface."*

2. Line 216-218. The sentence here doesn't make sense to me.

Response: We have removed the interpretation of the absence of cryo-EM structures of alpha-fetoprotein (AFP)'s homologous proteins, and the sentence has been revised to only demonstrate the fact: “*For the alpha-fetoprotein (AFP) with no symmetry, its other homologous structures were all determined by X-ray crystallography before*^{36,37}.”

3. One worry about the interaction of proteins with a surface (e.g. your UFG) is the possibility of strong interactions yielding preferred orientations. Clearly you did not have a big problem with this, in view of the beautiful reconstructions, but it would be nice to show in another supplementary figure the map of particle orientations.

Response: We thank the reviewer for this suggestion. We agree that the interaction of proteins with UFG may impose a risk of preferred orientation problem for some cases, which, however, could be alleviated by graphene modification. As suggested, we have made a figure to present the orientational distribution of these samples (Figure R1), which have been included in the revised manuscript (Extended Data Fig. 9).

Figure R1. Particle orientational distribution of hemoglobin (a), alpha-fetoprotein (b) and streptavidin (c).

Final Decision Letter:

Dear Hongwei,

I am pleased to inform you that your Article, "Uniform thin ice on ultraflat graphene for high-resolution cryo-EM", has now been accepted for publication in Nature Methods. Your paper is tentatively scheduled for publication in our December print issue, and will be published online prior to that. The received and accepted dates will be October 8, 2021 and October 20, 2022. This note is intended to let you know what to expect from us over the next month or so, and to let you know where to address any further questions.

Once your paper is typeset, you will receive an email with a link to choose the appropriate publishing options for your paper and our Author Services team will be in touch regarding any additional information that may be required.

Please note that *Nature Methods* is a Transformative Journal (TJ). Authors may publish their research with us through the traditional subscription access route or make their paper immediately open access through payment of an article-processing charge (APC). Authors will not be required to make a final decision about access to their article until it has been accepted. [Find out more about Transformative Journals](https://www.springernature.com/gp/open-research/transformative-journals)

Your paper will now be copyedited to ensure that it conforms to Nature Methods style. Once proofs are generated, they will be sent to you electronically and you will be asked to send a corrected version within 24 hours. It is extremely important that you let us know now whether you will be difficult to contact over the next month. If this is the case, we ask that you send us the contact information (email, phone and fax) of someone who will be able to check the proofs and deal with any last-minute problems.

If, when you receive your proof, you cannot meet the deadline, please inform us at rjsproduction@springernature.com immediately.

Once your manuscript is typeset and you have completed the appropriate grant of rights, you will receive a link to your electronic proof via email with a request to make any corrections within 48 hours. If, when you receive your proof, you cannot meet this deadline, please inform us at rjsproduction@springernature.com immediately.

Once your paper has been scheduled for online publication, the Nature press office will be in touch to confirm the details.

Once your paper has been scheduled for online publication, the Nature press office will be in touch to confirm the details.

Content is published online weekly on Mondays and Thursdays, and the embargo is set at 16:00 London time (GMT)/11:00 am US Eastern time (EST) on the day of publication. If you need to know the exact publication date or when the news embargo will be lifted, please contact our press office after you have submitted your proof corrections. Now is the time to inform your Public Relations or Press Office about your paper, as they might be interested in promoting its publication. This will allow them time to prepare an accurate and satisfactory press release. Include your manuscript tracking number NMETH-A47297B and the name of the journal, which they will need when they contact our office.

About one week before your paper is published online, we shall be distributing a press release to news organizations worldwide, which may include details of your work. We are happy for your institution or funding agency to prepare its own press release, but it must mention the embargo date and Nature Methods. Our Press Office will contact you closer to the time of publication, but if you or your Press Office have any inquiries in the meantime, please contact press@nature.com.

Nature Portfolio journals [encourage authors to share their step-by-step experimental protocols](https://www.nature.com/nature-research/editorial-policies/reporting-standards#protocols) on a protocol sharing platform of their choice. Nature Portfolio 's Protocol Exchange is a free-to-use and open resource for protocols; protocols deposited in Protocol Exchange are citable and can be linked from the published article. More details can found at www.nature.com/protocolexchange/about.

Please note that you and any of your coauthors will be able to order reprints and single copies of the issue containing your article through Nature Portfolio 's reprint website, which is located at <http://www.nature.com/reprints/author-reprints.html>. If there are any questions about reprints please send an email to author-reprints@nature.com and someone will assist you.

Best regards,
Arunima

Arunima Singh, Ph.D.
Senior Editor
Nature Methods